# Illusion or Algorithm? Investigating Memorization, Emergence, and Symbolic Processing in In-Context Learning

**Jingcheng Niu**                                    *jingcheng.niu@tu-darmstadt.de*
*UKP Lab, Technical University of Darmstadt*

**Subhabrata Dutta**                                *subhabrata.dutta@tu-darmstadt.de*
*UKP Lab, Technical University of Darmstadt*

**Ahmed Elshabrawy**                                *ahmed.elshabrawy@mbzuai.ac.ae*
*Mohamed bin Zayed University of Artificial Intelligence*

**Harish Tayyar Madabushi**                                    *htm43@bath.ac.uk*
*The University of Bath*

**Iryna Gurevych**                                    *iryna.gurevych@tu-darmstadt.de*
*UKP Lab, Technical University of Darmstadt*

**Reviewed on OpenReview:** *https://openreview.net/forum?id=10QqO1tM1H*

## Abstract

Large-scale Transformer language models (LMs) trained solely on next-token prediction with web-scale data can solve a wide range of tasks after seeing just a few examples. The mechanism behind this capability, known as in-context learning (ICL), remains both controversial and poorly understood. Some studies argue that it is merely the result of memorizing vast amounts of data, while others contend that it reflects a fundamental, symbolic algorithmic development in LMs. In this work, we introduce a suite of investigative tasks and a novel method to systematically investigate ICL by leveraging the full Pythia scaling suite, including interim checkpoints that capture progressively larger amount of training data. By carefully exploring ICL performance on downstream tasks and simultaneously conducting a mechanistic analysis of the residual stream's subspace, we demonstrate that ICL extends beyond mere "memorization" of the training corpus, yet does not amount to the implementation of an independent symbolic algorithm. Our results also clarify several aspects of ICL, including the influence of training dynamics, model capabilities, and elements of mechanistic interpretability. Overall, our work advances the understanding of ICL and its implications, offering model developers insights into potential improvements and providing AI security practitioners with a basis for more informed guidelines.[1]

## 1 Introduction

Large-scale Transformer based language models (LMs) trained exclusively on the next-token prediction task acquire the ability to perform a novel task based solely on a few demonstrations provided within the prompt, without any gradient updates—a phenomenon known as *in-context learning* (ICL; Brown et al., 2020). ICL, also referred to as few-shot learning, is a fundamental mechanism in LMs (Lampinen et al., 2025). Therefore, understanding the nature and generalization limits of ICL during the the next-token prediction pretraining process, is crucial. Additionally, understanding the nature of ICL is central to the safe and secure deployment of generative AI, as it has been argued that since increased scale alone is known to unlock new abilities in LMs, further scaling up has the potential to unlock latent hazardous abilities which carry important security

---

[1]The code and data of this work are publicly available online: `https://github.com/UKPLab/tmlr2025-icl-investigation`.

and safety implications. Simultaneously, Lu et al. (2024) provide strong correlations between the ability of a model to solve tasks in few-shot ICL and their generalizability on novel tasks upon instruction tuning. Therefore, understanding the fundamental mechanisms of ICL has broad implications, including safety, security, and the capabilities of user-facing, instruction-tuned LMs.

Despite its importance, the current understanding of ICL in LMs remains limited and is often clouded by contradictory claims and findings, unlike our understanding of traditional in-weight learning via gradient descent. Specifically, researchers continue to debate whether ICL is functionally equivalent to gradient descent (Von Oswald et al. (2023) *vs.* Deutch et al. (2024)), and whether ICL is primarily regurgitation data and patterns encountered during pretrained or genuine generalization—even to the point of implementing symbolic algorithms (Golchin et al. (2024) *vs.* Elhage et al. (2021)), *inter alia.* A significant majority of prior attempts to investigate the characteristics of ICL focus on either Transformers trained on purely synthetic data (Edelman et al., 2024; Singh et al., 2023) or mechanism construction within toy-sized Transformers (Elhage et al., 2021; Olsson et al., 2022); as such, it is hard to scale these findings to LMs trained on real-world language data. Simultaneously, there has been a discrepancy in the tasks considered to characterize ICL; e.g., Golchin et al. (2024) consider different NLP benchmark tasks (with the possibility of leakage), while Olsson et al. (2022) define ICL more narrowly as an LM's ability to replicate patterns from context. Intuitively, while the latter is a prerequisite for the former, several missing intermediary elements prevent a direct equivalence, leading to contradictory outcomes. This discrepancy within the scope of tasks to investigate language models' ability (or inability) to infer abstract rules from the context and perform symbolic operations based on those rules has continued beyond few-shot task solving. Prior works confirm the existence of symbolic computation (Prakash et al., 2025; Yang et al., 2025) as well as its brittleness (Shojaee et al., 2025; Mirzadeh et al., 2024). However, the tasks these two groups consider are of completely different types, and subsequently, one cannot draw any generalized conclusion.

Given this context, we investigate two fundamental research questions to characterize ICL in LMs trained on real-world, web-scale corpora:

1. **Generalization versus Memorization.** To what extent does ICL reflect an LM's capacity for true generalization, as opposed to large-scale memorization? In particular, does ICL merely regurgitate examples encountered during pre-training, or do LMs infer an underlying algorithm from the provided in-context examples?

2. **The Development of ICL Abilities and Mechanisms during Pre-training.** How can we characterize the development and manifestation of ICL *during pre-training*? Specifically, we investigate: a) whether the "emergence" of ICL is predictable, b) how it relates to model scale and pre-training data size, c) its interaction with task "difficulty," and d) the formation of internal mechanisms.

Towards a robust research design to answer these questions, we put forth a suite of novel tasks (§3) that are designed to isolate distinct aspects of ICL previously identified in the literature: in-context pattern matching *without* memorized concept associations from pre-training data (LSC, LSCG, WI and WC), in-context pattern matching *with* memorized concept associations (TT), and finally, in-context overriding of memorized concepts via counterfactuals (CF). We conduct experiments using the full Pythia scaling suite (including model of various scales *and* their interim checkpoints) to simultaneously evaluate the effects of model scale and amount of training data on ICL (Biderman et al., 2023). Our main contribution is that we identify three novel findings:

**Finding 1** We find that when provided with in-context examples, LMs trained on real-world corpora display a peculiar mix of generalized pattern-matching abilities on the one hand, and a dependence on pattern statistics on the other. Specifically, they *can* detect and follow patterns presented in-context using random token orderings—and therefore are unlikely to have memorized them during pre-training—refuting prior claims by Golchin et al.'s (2024). However, this pattern-learning ability declines when statistically rarer tokens are used to construct the patterns or when the pattern statistics in the examples are obfuscated, directly contradicting prior claims by Olsson et al.'s (2022), who describe ICL as symbolic algorithm execution. These two findings significantly refine our understanding of the mechanistic basis for ICL in LM, with implications to interpretability and circuit discovery (Wang et al., 2022; Conmy et al., 2023; Yu et al., 2024)

in LMs, as contemporary approaches often operate under the assumption that symbolic functions are being implemented within the models' internals.

**Finding 2**  Next, we observe that different aspects of ICL exhibit idiosyncratic developmental characteristics as training time and model size scale up. Specifically, we find that while the previously discussed performance gaps related to token frequency tend to stabilize with training, performance gaps due to task configurations (i.e., the same task with an "easier" *vs.* "harder" setup) typically persist. With regard to the development of ICL competence with increased pre-training, we observe that it is gradual and predictable on complex tasks, despite it appearing to be abrupt on "easier," pattern-based tasks, possibly due the lack of granularity in the pre-training checkpoints we experiment with. Taken together, these findings throw light on an explanation for why prior work has reported contradictory findings regarding the development of ICL competence. We provide the various aspects of models that must be considered in explaining the development of ICL competence—trying to understand ICL by focusing on just a few aspects in isolation is bound to fall short.

**Finding 3**  Upon investigating the model internals, we identify an aspect of the development that remains stable across different ICL characteristics. Specifically, we show that the dimensionality of the subspace of the residual stream (Elhage et al., 2021) used by the model to "carry" the answer token follows a consistent pattern: an initial rise in subspace allocation, corresponding to a "general" development of ICL competence, followed by a decay that saturates over training. Moreover, development of ICL competence through pretraining correlates with the model learning to use a common subspace across tasks. Both of these developments follow a predictable pattern, serving as a predictable reference variable for "occasionally abrupt" development of ICL competence. All of these findings bear importance to the notion of "emergent abilities of Large Language Models" introduced by Wei et al. (2022) and Ganguli et al. (2022) which are also related to ICL as described by Olsson et al. (2022); Lu et al. (2024): the predictability, or the lack thereof, of ICL has direct relevance to the possibility of other abilites being unexpectedly unlocked with further increase in scale, with significant implications to AI security.

## 2  Related Work

Our work is related to prior literature seeking to understand ICL in LMs. Here, we discuss the four broad categories of research in this direction: evidence of data contamination and memorization in LMs, empirical evaluation of what classes of tasks Transformer-based language models pre-trained on web-scale data can learn in-context, the learning dynamics of ICL and its relation to gradient descent, and finally the causal mediation analysis of internal components of the Transformer leading to ICL.

**ICL competence in language models.**  Brown et al. (2020) first empirically demonstrated that LMs, trained purely on next-token prediction (a.k.a. the *language modeling* task), can learn to perform a task by observing a few demonstrations in the input prompt. In this work, we coin the term **ICL competence** to refer to the general prowess of the LM in performing ICL, quantified by the model's performance on various ICL tasks. Our notion of ICL competence is much similar to Lampinen et al. (2025) who argue that ICL encapsulate a broad range of tasks beyond supervised few-shot learning. The number of studies showcasing what tasks LMs can do from ICL examples is too many to discuss individually within the scope of this paper; instead, we refer to the elaborate survey by Dong et al. (2024a). Min et al. (2022) attempted to disentangle the factors influencing an LM's ICL competence by perturbing the input-output space, subsequently claiming that LMs are often able to perform in-context classification correctly even when the labels are randomly permuted in the context. Wei et al. (2023) showcased that semantic priors of input-label mapping play a vital role in smaller LMs, while large LMs can perform ICL with semantically unrelated labels. Shi et al. (2023) explained this effect of scale on ICL competence by larger LMs' ability to process more in-context features. These works remain primarily focused on the static abilities of the trained model, without investigating how these abilities might have arisen via large-scale pretraining. In a somewhat orthogonal direction, Akyürek et al. (2024) investigated ICL competence through the lens of formal language modeling.

**Learning dynamics of ICL in Transformers.** A distinct line of attempts to understand ICL presents constructions of Transformer attention that can simulate gradient descent to argue that Transformers can implement certain classes of learning algorithms implicitly as ICL (Akyürek et al., 2022; Von Oswald et al., 2023). (Deutch et al., 2024) argued against this expressivity-based line of reasoning and pointed towards the distributional properties of training data that possibly dictate the development of ICL. This debate has been formalized to the question of whether ICL and gradient descent are equivalent or not (Dai et al., 2023; Mahdavi et al., 2024; Deutch et al., 2024). Prior work by Chan et al. (2022) has also pointed towards the data-distributional sources of ICL. Singh et al. (2023) demonstrated the transient nature of ICL in small Transformer models trained on synthetic data. (Chan et al., 2024) confirmed this transient nature and provided theoretical results that explain the learning dynamics of ICL. These works primarily identify the long-tailed distribution of the data as being the driver of ICL, whereas Bratulić et al. (2025) claimed that conceptual repetitions in the pretraining data are the primary determiner of ICL. Most of these prior investigations primarily rely on toy-sized Transformers and/or synthetic pretraining data. On the other hand, we focus on Transformers of a wide range of parameter scales, trained on real web-scale natural language data. To the best of our knowledge, Wang et al. (2024) comes as the only similar prior work in this direction; however, they sought to identify the dynamics of task recognition from ICL examples vs. task learning as they progress through pretraining.

**Causal mediation based interpretation of ICL.** Elhage et al. (2021) and Olsson et al. (2022) were among the first to demonstrate that LMs can consistently complete previously seen patterns in-context: `[A][B]...[A]→[B]`. They conjectured that certain internal model mechanisms (specifically, induction heads) form to perform this pattern-completion task as a symbolic algorithm. This hypothesis has since underpinned *mechanistic interpretability* research, which assumes LMs develop internal mechanisms akin to symbolic algorithms and tasks researchers with identifying them. It has inspired projects like Tracr (Lindner et al., 2023), which compiles symbolic algorithms into transformer weights, and circuit discovery methods (Wang et al., 2022; Conmy et al., 2023) to uncover them. Cho et al. (2024) recently identified the mechanistic circuitry of ICL as a three-step process. Upon analysis on synthetically trained Transformers, Park et al. (2024) found that the development of ICL is a juxtaposition of multiple algorithms and their completion, and any claim of 'singular' ICL mechanism should be taken with caution. In a somewhat less fine-grained method of interpreting ICL, the notion of function vectors or task vectors (Todd et al., 2023) has been proposed. Yin & Steinhardt (2025) identified attention heads that construct function vectors and show that 1) these heads are formed from induction heads upon further pretraining, and 2) these heads are the fundamental driver of ICL-specific circuits and not induction heads, as presumed by Elhage et al. (2021). More recently, Yang et al. (2025) hypothesise a three-stage symbolic processing circuit in LLMs for abstract rule-induction reasoning: abstraction, pattern induction, and retrieval. Wu et al. (2025) also observe a similar process for variable binding. In line with the school of mechanistic interpretability, our work validates the algorithmic nature of ICL, but puts significant limitations on the notion as well. While we seek to identify signals internal to the model that implicate ICL and its development with training and parameter scaling (Section 6), we do not focus on recovering the exact neural algorithmic implementation of ICL.

**Evidence of data contamination and memorization in LMs.** Is LMs' impressive performance merely a result of training data contamination? As contemporary LMs are trained on web-scale corpora, evaluation tasks may already appear in the pre-training dataset—or at least share a similar $n$-gram distribution with parts of it. Li & Flanigan (2024) found that tasks with datasets released prior to an LM's training data creation date perform surprisingly better, providing strong evidence that data contamination is commonplace in LMs. Li et al. (2024) offered a comprehensive report on data contamination, showing 1% to 45% contamination levels across various widely used benchmarks. In addition, several data contamination detection methods have been proposed (Golchin & Surdeanu, 2023; Dong et al., 2024b). Sainz et al. (2023) provided an excellent survey on data contamination. Most relevantly, Golchin et al. (2024) rightfully asked whether ICL is also influenced by data contamination, as they observed a very strong correlation between performance and memorization when ICL outperforms zero-shot learning. In this paper, we use the term **memorization** to refer to both the scenario where the model is able to solve a task because the exact prompt–answer sequence is present in pre-training data, *or* some sequence of text with similar $n$-gram statistics exists in the pre-training dataset. This is in contrast to the model genuinely acquiring the corresponding ability (e.g.,

Table 1: An overview of our six proposed tasks: LSC, LSCG, WC, WI, TT, and CF. In the examples, the prompt text is highlighted in green, and the expected target token is highlighted in orange.

| LITERAL-SEQUENCE-BASED TASKS CONSTRUCTED WITH RANDOM TOKENS |
| --- |

**Literal Sequence Copying** (LSC): We start with providing a systematic empirical evaluation of the literal sequence copying phenomenon identified by Olsson et al. (2022). In particular, we aim to test whether the model can generate `T` when prompted with `P* T R* P*`, where `T` is a target token and `P*` and `R*` are sequences of randomly sampled tokens of length |P| and |R|, respectively.

**Template:** `P* T R* P* T`     **Example:** Category 40 ids node struction     Yolk yes     Category 40 ids node     struction

                                  Pattern: `P*T`    Random gap: `R*`    Pattern prefix: `P*`    Target: `T`

**Task Configurations:** Pattern length (|P|): the number of tokens in the literal sequence pattern prefix (`P*`). Random gap length (|R|): the number of tokens in the random gap sequence (`R*`).

**Literal Sequence Copying with Random Gaps** (LSCG): We extend the pattern repeating task from Olsson et al.'s (2022) by introducing an additional challenge: a random gap within the pattern. Specifically, we modify the first occurrence of the pattern by inserting a gap of length |G|, denoted as (`U*`), while in the repeated instance, we introduce a different gap of the same length, labelled as (`V*`).

**Template:** `P* U* X T R* P* V* X T`     **Example:**

                             Pattern w/ Random Gap: `P*U*XT`       Random Gap: `R*`

                                Category 40 ids Garlic right node struction     Yolk yes Production

Category 40 ids total Content node       struction

Pattern w/ a Different Random Gap: `P*V*X`   Target: `T`

**Task Configurations:** Pattern length (|P|): the number of tokens in the literal sequence pattern prefix (`P*`). Random gap length (|R|): the number of tokens in the random gap sequence (`R*`). Random gap in pattern length (|G|): the number of tokens in the random gap sequence in the pattern with random gap (`U*` or `U*`).

**Word Content** (WC): A common way of using the ICL capability in LMs is to present the task as a classification task in the form of `prompt -> label`. We replicated this by presenting the model with a novel classification task of identifying whether a token (representing a feature) appears in the prompt sequence or not. We conceptualize this task as the basic form of common ICL applications such as sentiment classification (whether concepts with positive or negative features exists in the input). In particular, given a list of $k$ targets (`T1, ..., Tk`), output the corresponding label `Li` for each target `Ti` in the sequence. Each target-label pair is demonstrated $n$ times.

**Template:** `X* Ti X* -> Li;`     **Example:** 40 ids node -> gluten;... ids Tim yes -> gluten

**Task Configurations:** Number of features (|F|): the number of tokens to identify in sequence. Number of labels (|L|): the number classification labels. Number of distractors (|D|): the number of distractor tokens in the classification sequence.

**Word Index** (WI): Next, we seek to check the ability of the LM to identify sequence ordering in context by learning to copy a token from a particular position to the output. Given the model with $d$ different demonstrations of `S1 ... Sn -> Si;`, we examine whether it can successfully complete `T1 ... Tn -> Ti`.

**Template:** `S1 S2 ... Sn -> Si`     **Example:** 40 ids node -> ids; Tim crane yes -> crane; ... total mark Yolk -> mark

**Task Configurations:** Sequence length (|S|): the number of tokens in a sequence. Target index ($i$): the index of the target token.

| IN-CONTEXT PATTERN MATCHING & OVERRIDING OF PARAMETRIZED INFORMATION |
| --- |

**Token Translation** (TT): The tasks mentioned hitherto are all constructed using randomly sampled token sequences and do not require any form of *world knowledge* acquired via pre-training. To address this, we introduce a task where the model needs to utilize pre-trained knowledge in addition to pattern recognition from the context. Specifically, we construct a token-level translation task—identified by Brown et al. (2020) as one of the three ICL abilities in GPT-3 that took the most training to develop. For this task, we sample 200 common words in English, German, Spanish, and Italian. After presenting the model with $n$ pairs of translation examples (e.g., `cat -> Katze;`), we ask it to translate a new word (e.g., `dog ->`) and observe whether the LM correctly outputs Hund.

**Template:** `<EN> -> <DE>`     **Example:** cat -> Katze; owl -> Eule; dog -> Hund

**Counterfactual** (CF): Lastly, we present a task where counterfactual in-context information *overrides* pre-trained knowledge. Specifically, we require the model to perform the following task: if we switch the capitals of `A` and `B`, then `A`'s capital is `CB` and `B`'s capital is __. We then check whether the model outputs `CA`. Completing this task requires strong generalization, as the answer `CA` never appears in the context. Moreover, the phrase "`B`'s capital is `CA`" is counterfactual and should therefore be surprising to the model. Therefore, we use this task to test whether the intended override of information has taken place.

**Example:** If we switch the capital of Canada and Germany, then Canada's capital is Berlin and Germany's capital is Ottawa

the ability to understand social situations if it is able to solve a QA task about social situations such as SocialIQA).

# 3 Experimental Setup: Tasks, Data and Models

**ICL as in-context distribution recovery.** Previous work primarily defines ICL as the ability to learn an input—label mapping based on explicit in-context samples, with the only exception being Elhage et al.

(2021), who investigated literal sequence copying as a precursor to ICL. In this work, we argue that any next-token prediction that requires the model to recover a token distribution from the context and utilize it (with or without another token distribution memorized from the pre-training) qualifies as ICL—a notion of ICL very similar to Xie et al. (2021). This definition allows us to define pattern-matching tasks similar to Olsson et al. (2022), standard ICL examples of input-label pairs, as well as counterfactual reasoning with in-context facts under the broad umbrella of ICL.

**Limitations with existing benchmarks.** To the best of our knowledge, all prior works evaluating ICL (Conneau & Kiela, 2018; Hernandez et al., 2023; Todd et al., 2023) have used tasks that require some form of factual, common-sense, or linguistic knowledge and therefore may be subject to data contamination. Since these web-scale training datasets often comprise billions or even trillions of tokens, we cannot completely rule out the possibility—even when the pre-training dataset is fully accessible (e.g., Biderman et al., 2023; Groeneveld et al., 2024). Additionally, with real-world, natural-language tasks, it is practically impossible to control task configurations (e.g., feature sparsity, length dependence, token frequency statistics, reliance on parametrized knowledge, etc.), and as a result, one cannot reliably correlate model scale (training time and parameter size) with specific task characteristics.

**Proposed task suit.** To address the limitations of using existing benchmark data as explained earlier, in Table 1, we propose a novel suite of tasks constructed from various patterns using randomly sampled tokens, including literal sequence copying (LSC), literal sequence copying with random gaps (LSCG), word content (WC) and word index (WI). These tasks satisfy the notion of ICL as recovering and inferring from token distributions in-context. Since these tasks appear as sequences of random tokens, it is functionally impossible for them to exist in the pre-training dataset. Therefore, if an LM can perform these tasks, it cannot be due to memorization—some form of pattern generalization must be at play. Moreover, we introduce two tasks focused on associating and overriding parameterized information using the ICL pattern identification capability: we use token translation[2] (TT) to investigate how patterns are combined with parameterized information, and counterfactuals (CF) to observe how this parameterized information can be overridden by counterfactual prompt settings. Since these tasks require specific parameterized information, we cannot guarantee that it does not appear in the pre-training dataset; however, this is acceptable, as we already include pattern-based tasks that do not rely on such information and serve as a cleaner test of ICL capabilities. We assume that they are "harder" than the pattern-based tasks, as they involve an additional layer of combining or overriding information.

We conduct our experiments using Pythia (Biderman et al., 2023) and LLaMA (Touvron et al., 2023) models. Unless specified otherwise, we randomly sample tokens from the Brown (Francis & Kucera, 1979) corpus. For each task, we run 1,000 examples and report task accuracy based on whether the LM selects the target token as the most probable output.

## 4   Generalizability of ICL

Using the experimental setup defined above, we aim to answer our first research question. Specifically, we investigate whether ICL in LMs trained on large-scale natural language corpora is a purely algorithmic ability or memorization-driven ability. Since we aim to compare algorithmic and memorization, we make use of the pattern-based tasks that do not require any pre-trained knowledge (i.e., LSC, LSCG, WI, and WC).

### 4.1   ICL is not Pure Memorization

Table 2 shows an excerpt of the Pythia models' performance on the pattern-based tasks. When we zoom in to the LSC task, we can observe that **every model with a size larger than 400M parameters achieves near-perfect performance.**[3] We note that all tokens used in the task are randomly sampled, and therefore, this task is *unseen* to LMs which are pre-trained on natural language text. Thus, *if the model*

---

[2]For the TT task, we sample 200 common words across four languages: English, German, Spanish, and Italian. The sampling process is detailed in Appendix A.

[3]Full results from more settings and on the LLaMA models are presented in Appendix B, which also supports these findings.

Table 2: Selected Models' Performance across our proposed tasks.

| Setting | | | Pythia Models Accuracy (%) | | | | | | | | | | LLaMA Models Accuracy (%) | | | | | |
|---|---|---|---|---|---|---|---|---|---|---|---|---|---|---|---|---|---|---|
| | | | 14M | 31M | 70M | 160M | 410M | 1B | 1.4B | 2.8B | 6.9B | 12B | 3.2-1B | 3.2-1B-I | 3.2-3B | 3.2-3B-I | 3.1-8B | 3.1-8B-I |
| **\|P\|** | **\|R\|** | | *Literal Sequence Copying* (LSC) | | | | | | | | | | | | | | | |
| 5 | 5 | | 2.1 | 1.7 | 6.5 | 45.4 | 96.3 | 95.8 | 93.0 | 95.1 | 97.7 | 98.4 | 99.7 | 98.4 | 100.0 | 99.3 | 100.0 | 99.9 |
| 10 | 10 | | 3.0 | 1.6 | 4.8 | 51.8 | 99.3 | 98.6 | 98.7 | 99.7 | 99.8 | 99.9 | 99.6 | 100.0 | 100.0 | 99.8 | 100.0 | 100.0 |
| 20 | 20 | | 2.1 | 7.2 | 2.4 | 60.0 | 99.2 | 99.7 | 99.7 | 99.8 | 99.8 | 100.0 | 99.8 | 99.9 | 100.0 | 99.9 | 100.0 | 100.0 |
| **\|P\|** | **\|R\|** | **\|G\|** | *Literal Sequence Copying with Random Gaps* (LSCG) | | | | | | | | | | | | | | | |
| 10 | 10 | 0 | 2.4 | 2.6 | 4.0 | 55.4 | 98.4 | 99.3 | 99.0 | 99.6 | 99.8 | 99.9 | 100.0 | 99.9 | 100.0 | 99.9 | 100.0 | 100.0 |
| 10 | 10 | 5 | 1.5 | 0.9 | 0.6 | 17.9 | 64.9 | 71.9 | 79.2 | 86.5 | 74.1 | 84.4 | 96.6 | 92.0 | 97.8 | 87.0 | 97.3 | 93.2 |
| 10 | 10 | 10 | 1.1 | 0.9 | 0.2 | 8.7 | 45.2 | 55.6 | 61.9 | 76.1 | 58.9 | 70.7 | 88.7 | 71.0 | 93.7 | 55.4 | 93.7 | 73.0 |
| 20 | 20 | 0 | 1.8 | 5.9 | 1.9 | 55.7 | 99.6 | 99.6 | 99.7 | 99.9 | 99.9 | 100.0 | 100.0 | 100.0 | 100.0 | 99.8 | 100.0 | 100.0 |
| 20 | 20 | 5 | 1.2 | 1.0 | 0.1 | 8.9 | 61.2 | 79.1 | 78.6 | 87.6 | 65.7 | 80.2 | 95.6 | 93.5 | 97.9 | 91.7 | 96.6 | 95.8 |
| 20 | 20 | 10 | 0.4 | 0.2 | 0.1 | 5.6 | 43.2 | 67.9 | 70.3 | 73.4 | 50.4 | 64.8 | 92.7 | 87.0 | 96.3 | 75.2 | 95.5 | 85.3 |
| **\|S\|** | ***i*** | | *Word Index* (WI) | | | | | | | | | | | | | | | |
| 4 | 0 | | 0.0 | 0.1 | 0.1 | 21.4 | 61.9 | 72.3 | 84.7 | 85.5 | 68.0 | 77.8 | 34.9 | 88.6 | 47.3 | 70.0 | 87.8 | 90.4 |
| 4 | 1 | | 0.0 | 0.1 | 0.0 | 2.8 | 12.8 | 15.7 | 21.5 | 18.4 | 24.2 | 22.4 | 12.4 | 19.7 | 26.4 | 19.5 | 40.0 | 30.0 |
| 4 | 2 | | 0.0 | 0.0 | 0.0 | 2.4 | 23.3 | 25.9 | 27.7 | 23.6 | 29.9 | 29.7 | 23.6 | 18.8 | 28.6 | 15.4 | 33.8 | 23.3 |
| 4 | 3 | | 0.0 | 0.0 | 0.0 | 16.2 | 62.6 | 78.0 | 78.9 | 89.3 | 84.9 | 90.1 | 93.5 | 96.8 | 97.0 | 98.3 | 99.1 | 98.6 |
| **\|F\|** | **\|L\|** | **\|D\|** | *Word Content* (WC) | | | | | | | | | | | | | | | |
| 2 | 2 | 0 | 6.5 | 12.7 | 25.5 | 55.2 | 97.5 | 96.7 | 97.5 | 96.4 | 99.4 | 99.4 | 99.3 | 93.9 | 99.0 | 97.9 | 100.0 | 100.0 |
| 2 | 2 | 1 | 4.4 | 10.3 | 14.3 | 46.2 | 92.4 | 95.5 | 94.9 | 91.8 | 98.3 | 98.6 | 97.9 | 88.3 | 90.7 | 97.8 | 100.0 | 99.7 |
| 2 | 2 | 2 | 4.7 | 10.0 | 13.7 | 40.4 | 87.6 | 91.1 | 88.9 | 88.0 | 93.3 | 95.0 | 95.5 | 82.8 | 84.2 | 95.9 | 99.1 | 99.4 |
| 2 | 2 | 5 | 5.8 | 8.1 | 16.0 | 40.9 | 75.1 | 77.9 | 75.7 | 71.5 | 81.5 | 82.3 | 77.5 | 73.0 | 69.9 | 75.0 | 90.7 | 93.9 |
| 2 | 2 | 10 | 5.6 | 9.9 | 18.6 | 37.1 | 65.7 | 65.0 | 66.9 | 60.5 | 68.3 | 66.9 | 57.4 | 59.5 | 58.4 | 57.8 | 71.8 | 75.3 |
| Setting | | | *Token Translation* (TT) | | | | | | | | | | | | | | | |
| EN → DE | | | 0.1 | 0.1 | 0.7 | 2.9 | 22.2 | 32.0 | 54.9 | 70.9 | 77.6 | 82.5 | 76.40 | 79.00 | 87.90 | 85.50 | 90.20 | 89.70 |
| EN → FR | | | 0.1 | 0.1 | 0.8 | 5.5 | 29.3 | 36.2 | 63.2 | 75.2 | 85.5 | 85.1 | 83.30 | 79.00 | 91.30 | 89.20 | 93.10 | 93.20 |
| EN → ES | | | 0.5 | 0.2 | 0.4 | 4.3 | 30.8 | 40.7 | 66.7 | 78.3 | 82.2 | 86.9 | 84.80 | 77.50 | 87.90 | 85.20 | 91.80 | 91.00 |
| EN → IT | | | 0.2 | 0.0 | 0.7 | 4.0 | 20.3 | 33.3 | 57.3 | 74.7 | 80.0 | 84.9 | 82.90 | 74.30 | 89.00 | 85.90 | 92.90 | 93.90 |
| | | | *Counterfactual* (CF) | | | | | | | | | | | | | | | |
| CF | | | 0.0 | 0.0 | 0.1 | 2.5 | 7.1 | 33.3 | 22.9 | 72.6 | 85.0 | 87.9 | 73.0 | 83.0 | 95.9 | 95.9 | 98.4 | 97.1 |

*only relies on token co-occurrence statistics and does not possess any capability to generalize in the algorithmic and structural manner we detail above, the model cannot achieve the high performance demonstrated.* Therefore, *consistant with* Golchin et al. (2024), we confirm that certain aspects of ICL are independent of memorization. While *some* instances of ICL may arise from memorization, this pattern-matching and copying capability clearly goes beyond mere memorization. Our findings, in this context of large LMs trained on natural language corpora, reproduce Olsson et al. (2022)'s findings on toy Transformer models. However, we emphasize that this does not imply that ICL is a symbolic algorithm that works irrespective of the task structure or token statistics. To evaluate this, we next explore the effects of unigram token frequency and task configurations—designed to loosely control the notion of the complexity or difficulty of the tasks—on the ICL performance.

## 4.2 ICL Performance Strongly Correlates with Token Frequency

But, does this ability of LMs to generalize beyond memorization mean the models have learned to carry out some symbolic algorithm, as Olsson et al. (2022) originally posited? Here, we present evidence that LMs' pattern-repeating behavior is still dependent on token co-occurrence statistics. Specifically, LMs' pattern-repeating performance strongly correlates with the frequency of the tokens that make up the patterns in the task.

Nevertheless, calculating the frequency of individual tokens in web-scale pre-training data requires a prohibitively large amount of computation resource, and many LMs—such as the GPT and LLaMA families—have not disclosed their pre-training datasets. Therefore, we apply a novel method that uses the *tokenizer index* as a proxy for the unigram frequency of a token in the tokenized corpus. Most modern LMs, including Pythia and LLaMA (the two types of models used in this work), rely on tokenizers based on the byte pair encoding (BPE; Sennrich et al., 2016) algorithm, where the most common contiguous character sequences are

Table 3: Pearson correlation between performance and token indices shows a strong negative correlation that is statistically significant across all models. This confirms our finding that model performance decreases as patterns contain tokens sampled from ranges with larger indices—i.e., tokens that appeared less frequently during pre-training.

|  | 14M | 31M | 70M | 160M | 410M | 1B | 1.4B | 2.8B | 6.9B | 12B |
|---|---|---|---|---|---|---|---|---|---|---|
| Pearson $r$ | -0.8511 | -0.6722 | -0.6540 | -0.9126 | -0.7392 | -0.6571 | -0.7010 | -0.7958 | -0.6967 | -0.7591 |
| $p$-value | 9.44e-15 | 1.23e-07 | 3.47e-07 | 6.99e-20 | 1.31e-09 | 2.93e-07 | 2.03e-08 | 8.26e-12 | 2.71e-08 | 2.61e-10 |

merged into tokens first. As such, tokens with smaller indices (merged earlier in the process) are likely to be more frequent than those with larger indices.[4] In this experiment, we test whether LMs perform differently on tasks involving tokens with different indices.

Here, we construct the LSC task[5] using two types of tokens. First, we use the same frequent, full-word tokens sampled from the Brown corpus as in the previous experiment. Second, we divide the tokenizer's index range into 1000-token intervals (e.g., 10k–11k and 30k–31k) and directly sample tokens from those intervals. Figure 1 shows how the LMs perform on the LSC task with tokens sampled from different ranges. Overall, we observe a general trend: models perform better on more frequent tokens (lower token indices) and worse on less frequent ones (higher token indices). This trend is statistically significant. In Table 3, we run a Pearson correlation test and find that there is a strong, negative correlation between the LM's LSC performance and the token indices across across every model size. Moreover, all LMs perform better on the full-word token control group, achieving 98.6–99.9% LSC accuracy, compared to every tokenizer index range. Results for Llama models are shown in Appendix C, results for

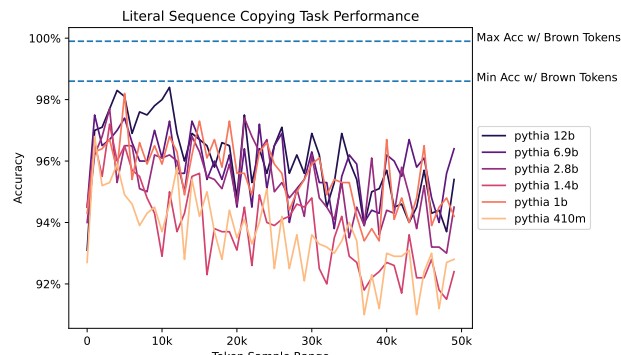

Figure 1: LM performance on the LSC task deteriorates as tokens are sampled from ranges with larger indices. The figure shows the final checkpoint performance of various sufficiently large Pythia models (410M+) on the pattern repeating task, with tokens drawn from different index ranges. The maximum and minimum performance on the same task but with Brown Corpus tokens are highlighted by the two vertical lines.

other literal-based ICL tasks (LSCG, WC, WI) are shown in Appendix D and they support the same conclusion.

This surprising finding directly refutes Olsson et al.'s (2022) particular claim that LMs can identify and follow literal patterns `[A][B]...[A]→[B]` "regardless of what A and B are," (Olsson et al., 2022) once induction heads are formed. If the tokens appear less frequently in the pre-training corpus, the LSC performance is less robust. This finding clearly shows that the so-called *literal sequence copying* behavior is significantly dependent on token statistics. Moreover, Olsson et al. (2022) quoted their discovery of LMs' ability to perform literal sequence copying as a "surprising and unwanted off-distribution generalization" (Olsson et al., 2022), and argued that this generalization ability should warrant concern in the AI security community. We agree that this literal sequence copying ability—and other similar abilities we tested in the previous subsection— shows that LMs demonstrate some level of off-domain generalization beyond memorization through ICL. However, this off-domain generalization ability has its limits, as it is still significantly affected by simple token occurrence statistics. Importantly, When discussing AI security (i.e., obtaining novel reasoning abilities based on the prompt alone), we think researchers should take these limitations into account.

---

[4]The BPE tokenizer is often trained on a different dataset, and the token indices do not map directly to unigram frequencies. Moreover, "open-weight" LMs such as Llama do not release their pre-training datasets, making direct comparison practically impossible. Therefore, we sampled 5,000 tokens from the Pythia tokenizer, counted their token frequencies on 250,000 documents from the Pile, and computed the correlation between tokenizer index and token count. We observed a Spearman $r = -0.746$ and a Pearson $r = -0.701$, both with $p < 10^{-100}$. This suggests that the tokenizer index is a reasonable and effective approximation of unigram token frequency statistics.

[5]We set both the pattern length and the gap length to 10. Other settings show similar results.

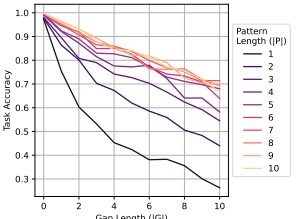 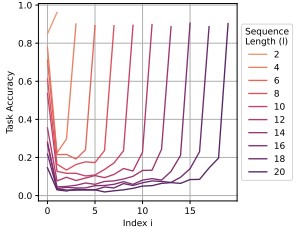 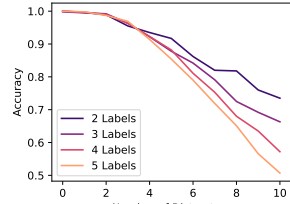 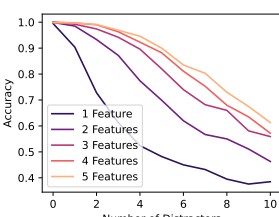

(a) LSCG performance varies over gap length ($|G|$) and pattern length ($|P|$).

(b) Word Index (WI) performance follows a U-shape trend with respect to sequence length and index.

(c) Word Content (WC) performance intuitively depends on task settings. Increasing the number of distractors, label classes, and features raises task difficulty, which in turn lowers accuracy.

Figure 2: ICL performance varies across tasks and configurations designed to control the "difficulty" of tasks.

Lastly, we note that Olsson et al. (2022) discuss the "fuzziness" of LMs' LSC ability. However, they approached the topic from a different perspective than our findings. They identified that LSC also applies to `[A*][B*]...[A][B]`—"the fuzzy nearest neighbor match" or "find something similar early in the sequence and complete the sequence in analogy" (Olsson et al., 2022). This type of fuzziness does not challenge their hypothesis that induction heads form an algorithmic behavior: when identifying a repeating pattern (at token `[A]`), the induction head copies the previous (representation of `[B*]`) into the residual stream and generates the similar `[B]` as the next token. This hypothesis cannot explain the difference of the model at handling different types of tokens with different frequencies.

### 4.3 ICL Performance Varies with Task Configuration Choices

Recall that Table 2 shows how LMs perform across all the literal-sequence-based tasks we constructed. While LMs achieve near-perfect performance on LSC, arguably the simplest task, their performance drops substantially when we modify the task configurations to make the tasks more challenging or apply them to other proposed tasks. Therefore, these configurations give us a unique way to control the "solvability" or "difficulty[6]" of the ICL tasks, giving us a more fine-grained way to study the models' ICL competence. Figure 2 shows how `Pythia-12B`'s performance changes with different task configurations.

For LSCG, we find that increasing the gap length ($|G|$) or decreasing the pattern length ($|P|$) makes the task harder and leads to lower performance (Figure 2a). In other words, it is harder for the model to identify and copy the literal sequence when the random gap is longer or the sequence is shorter.

On WI, ICL performance follows a U-shaped trend with respect to the selected index $i$, as illustrated in Figure 2b. Task accuracy is highest at the two endpoints and much lower in the middle. Interestingly, performance is better at the right endpoint (end) than at the left endpoint (beginning). Increasing the sequence length $L$ also makes the task harder—as shown in the figure, the darker lines fall under the lighter ones, and the left endpoint performance for shorter sequences is also higher than that for longer sequences.

Lastly, for WC, task performance is affected by the number of distractors, the number of features (i.e., how many words to identify), and the label classes. Figure 2c shows how model performance varies across these three configurations.

Task performance results for other Pythia models (listed in Appendix B) confirm the same findings.

In sum, we argue that a true "symbolic algorithm" should be robust to minor changes in task structure. However, we find that task performance is related to three simple factors: the gap length in the LSCG task, the sequence length in the WI task, and the number of distractors or features in the WC task.

---

[6]We use the terms *solvability* and *difficulty* loosely to refer to general LM performance (accuracy) on the task. We are not attempting to validate task complexity in a cognitive science sense, which is outside the scope of this paper.

### 4.4 Memorization, Symbolic Algorithms, or Generalization: What Is the Nature of ICL?

Our experiments offer a comprehensive picture of the nature of ICL in relation to *memorization*, the implementation of some *symbolic algorithm*, and *generalization*. In summary, we cannot dismiss ICL as mere memorization of the pre-training dataset—ICL is indeed a truly novel capability and not just a byproduct of data leakage or improper evaluation. Yet, the hypothesis that LMs conduct ICL by following stringent symbolic algorithms also faces some challenges, as LMs' ICL performance tends to vary across (1) task configuration and (2) token frequency, indicating that the capability is still statistical and generalization-based in nature.

Importantly, we demonstrate that ICL does not provide unconstrained off-distribution or out-of-domain generalization, as highlighted by our surprising finding that the model's LSC performance decreases as the literal patterns are composed of less frequent tokens. This finding highlights that notions of *scope*, *domain*, and *distribution* are nuanced and multifaceted when discussing the threat posed by this novel ICL capability to AI security. Olsson et al. (2022) considered ICL a demonstration of the possibility to fundamentally alter the behavior of an LM during inference without further training; and described it as off-distribution generalization that could be surprising and unwanted. Our finding suggests that an LM's ICL capability is still tied to pre-training and token co-occurrence statistics. While ICL demonstrates off-distribution generalization in the task or functional domain, the generalization has limits in the token domain. We hope our findings prompt a more nuanced look into the implications of ICL for AI security.

Importantly, we do not dispute recent research on data contamination and LM memorization (Golchin & Surdeanu, 2023; Li & Flanigan, 2024; Li et al., 2024, *inter alia*), since both effects can occur simultaneously. They rightfully pointed out that what many prior works called ICL is likely just sophisticated memorization. However, our research shows that ICL is a genuinely novel capability that LMs possess, and that does not contradict their findings. In fact, their work inspired us to develop all of our literal-pattern-based tasks and reminded us of the importance of proper experimental setup and robust evaluation regimes.

Finally, we address a legitimate concern that these observations may result from insufficient scaling, either in training time or model size—a point we investigate further in the next section. Looking at the final checkpoint alone does not prove that these patterns are fundamental to ICL. With more training or larger model sizes, they might disappear. Moreover, there is a recent popular notion that LM ability, including ICL, develops in an abrupt and unpredictable manner (Wei et al., 2022; Olsson et al., 2022). So, in the next section, we extend our analysis across the full pre-training process, using the checkpoints collected by the Pythia project.

## 5 Development of ICL & Scaling

In this section, we characterize the development of ICL competence by examining checkpoints across the entire pre-training process. We begin by examining the two factors affecting LMs' ICL competence, identified in the previous section: token frequency (tokenizer index) and task settings. Our investigation shows that these characteristics remain consistent as models scale in training time and size—i.e., they cannot be changed by further scaling and are therefore more likely to be inherent to ICL. We then extend the discussion to characterize the overall performance of ICL and find that its development can be gradual and predictable.

### 5.1 In-distribution ICL Develops First, and the In- and Off-distribution Gap Stabilizes Over Time

Recall that we previously identified that LM performance on the LSC task differs when sampling from different random token ranges. Here, we investigate how this performance difference develops across training steps. Specifically, since Pythia provides early training checkpoints in the log space and thereafter every 1000 steps $(1, 2, 4, \ldots, 512, 1000, 2000 \ldots, 143000)$, we leverage these checkpoints to analyze the LSC performance difference on tokens with different frequencies. Is this gap in in- and off-distribution generalization persistent across the training process? Also, to revisit the concern raised in the previous section: is the nuance we observe in generalizability fundamental to the nature of ICL, or just a fluke of limited scaling?[7]

---

[7]Since Pythia models smaller than 410M showed no competence on the LSC task, we do not include them in this study.

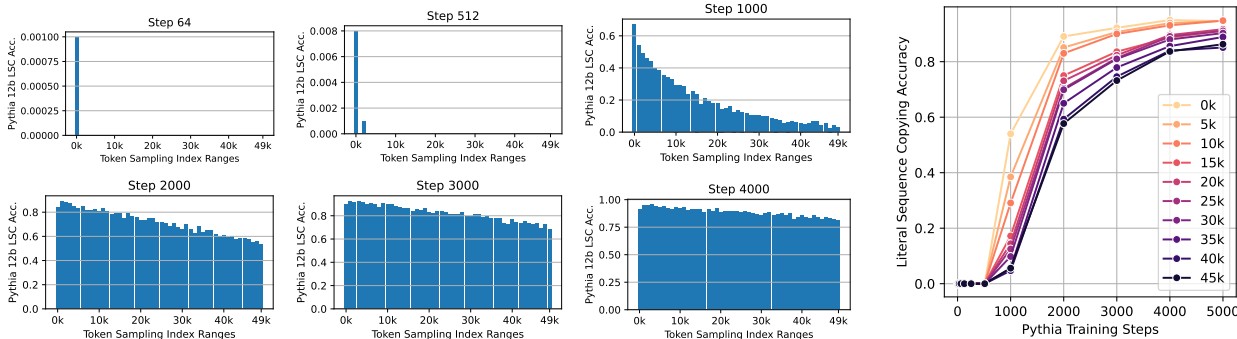

(a) Breakdown of LSC performance across token indices in several early Pythia checkpoints. The LMs' LSC capability develops first on more frequent tokens (those sampled from lower indices) and gradually converges to a stable level.

(b) An overview of early stage LSC ability development. Line colors indicate token index ranges.

Figure 3: Literal sequence copying (LSC) ability develops differently for in- and off-distribution generalization in the early stages of pre-training. In-distribution generalization develops earlier and more rapidly, whereas off-distribution generalization tends to develop later and more slowly.

**Early stages of pre-training.** We start by investigating the early periods of the pre-training process. We find that LMs' in- and off-distribution generalization capabilities develop at different paces. Figure 3 shows how `Pythia-12b` develops the LSC. Specifically, in-distribution generalization ability develops earlier and faster, as we observe that LSC with more frequent tokens improves more quickly compared to those with less frequent tokens. From step 1 to 512, the model shows no meaningful LSC capability; i.e., performance remains close to 0 across all these checkpoints. However, at earlier steps (e.g., steps 64 and 512 in Figure 3a), the model achieves some accuracy (0.1% to 0.8%) on the most frequent index ranges, while showing no accuracy on the rest. At step 1000, some meaningful LSC capability begins to appear, with performance following a long-tailed distribution. By steps 2000 to 4000, the disparity in performance across token index ranges diminishes, though it does not vanish entirely. In summary, these observations indicate that in-distribution generalization develops earlier and more rapidly during pre-training compared to the harder off-distribution generalization ability.

Moreover, we evaluate using continuous metrics (particularly, log probability[8]) in Figure 4. As Schaeffer et al. (2023) pointed out, accuracy—as a discrete measure—can obscure gradual improvements, fail to reflect meaningful changes in model behavior, and even lead to misleading conclusions. In our case of LSC, using a continuous metric does reveal the same dispersion of in- and off-domain development. The model's performance shows a clear stratification by token frequencies (tokenizer indices), with in-distribution LSC ability on more frequent tokens developing earlier and faster, and *vice versa* for off-distribution ability on rare ones.

It is worth mentioning that we observe an interesting, abrupt development in the models' LSC ability between step 512 and step 1000. With the more discrete accuracy measure, meaningful performance first begins to appear between these two time steps; with the continuous log probability measure, on the other hand, we see a sudden jump in log probability be-

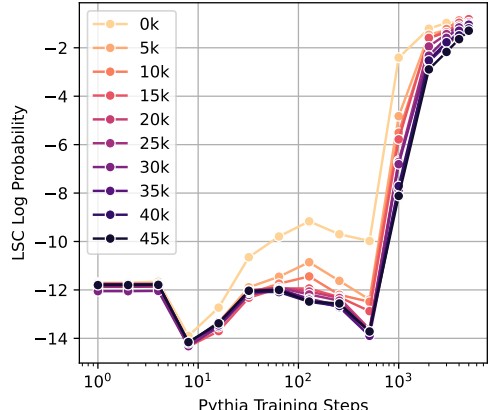

Figure 4: The early development of `Pythia-12b`'s LSC ability measured in log probability, a continuous metric. The result supports the same findings.

tween these two points, preceded by a period of decline in log probability. This development resembles a

---

[8]All of our tasks have exactly one correct answer token, and we follow Schaeffer et al.'s (2023) suggestion to apply the per-token log probability measure for this task setup.

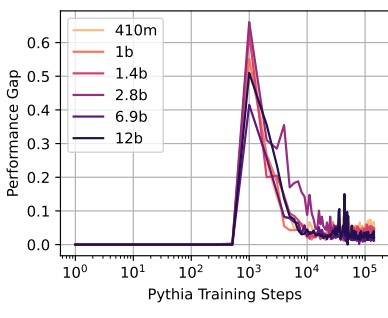 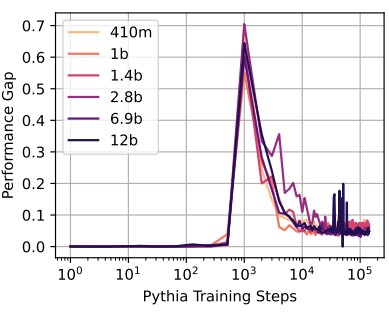 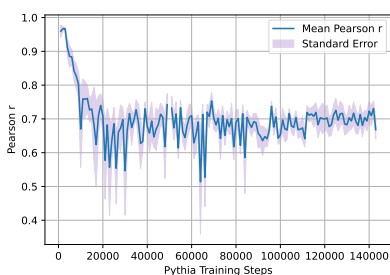

First *vs.* Last.  Best *vs.* Worst.

(a) Performance gap throughout the pre-training process across model sizes. The gap is always positive, with the largest difference appearing when the pattern-repeating behavior first appears (step 1000), after which it decreases to a stable level. Training step ($x$-axis) is shown on a logarithmic scale.

(b) Pearson correlation between LSC performance and token indices (aggregated across all model sizes). The shaded blue area indicates the error range. We can observe the correlation stabilize with more training.

Figure 5: The dispersion between in- and off-distribution generalization ability. For all model sizes, we observe that in-distribution generalization develops earlier and faster. The gap decreases with more training but stabilizes at a stable, positive level.

phase change, similar to the one identified by Olsson et al. (2022), and thus we conjecture that this period corresponds to the same phase they identified, during which the attention heads form.

**All model sizes & the complete pre-training process.**  This phenomenon, where in-distribution ICL develops first and off-distribution ICL develops later, is observed across all Pythia model sizes. Figure 5a shows how the Pythia models' performance gap, measured either between the first and last token index ranges ($Accuracy_{0k} - Accuracy_{50k}$) or between the best- and worst-performing index ranges ($Accuracy_{\max} - Accuracy_{\min}$), demonstrates the same trend. There is a spike in performance between step 512 and 1000, when meaningful LSC ability first appears, and the gap gradually decreases to a stable, positive level. The sizes of the performance gaps are, overall, very similar across model sizes, suggesting that this dispersion between in- and out-distribution generalization ability is independent of model size. This characteristic of ICL is not likely to be bridged by scaling up the size of the models.

Nevertheless, at each time step for each model, we gather 51 performance datapoints for each index range (0k–50k). Although our previous results demonstrated that the trend persists across model sizes, they did not present the full picture across all token ranges. Therefore, in Figure 5b, we show how the Pearson correlation between the model's LSC performance and the index ranges evolves across all training steps. This more comprehensive evaluation again affirms our main observation: we see a very strong Pearson correlation ($\sim$0.95) when LSC first develops at early steps ($\sim$1000 training steps); the correlation then drops to a still-strong value of $\sim$0.7 at around 20,000 steps, and stabilizes near 0.7 through to the end.

**Summary of findings.**  Together, we conclude that in-distribution ICL ability — as represented by the LSC task on various token index ranges — develops earlier and faster in the pre-training process compared to off-distribution generalization. Using our novel method of using token index to establish a spectrum from in- to off-distribution data, we can now study this more nuanced level of generalization in LMs with respect to ICL, and find that generalization speed itself also follows the same spectrum, slowing down progressively as the data shifts further off-distribution.

Furthermore, we investigate all sufficiently large Pythia models (410M+) and analyze the training dynamics across the entire pre-training process. We first find that the training dynamics are largely consistent across all model sizes. Thus, the behavior is not likely an artifact of the model being too small, but rather something more fundamental to LMs' ability to generalize over the structure of the literal sequence. Second, we discover that although the gap decreases after the very early stages of pre-training, it stabilizes at a roughly constant level around 20,000 training steps. This shows that the dispersion is also not caused by the model being

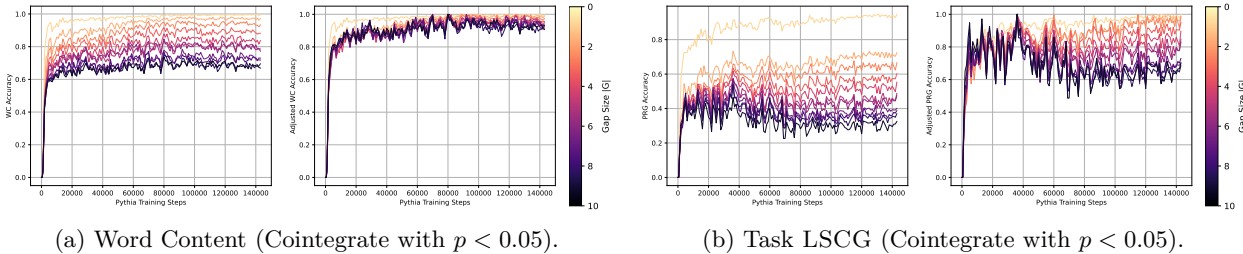

(a) Word Content (Cointegrate with $p < 0.05$).     (b) Task LSCG (Cointegrate with $p < 0.05$).

Figure 6: ICL competence development across task configurations.

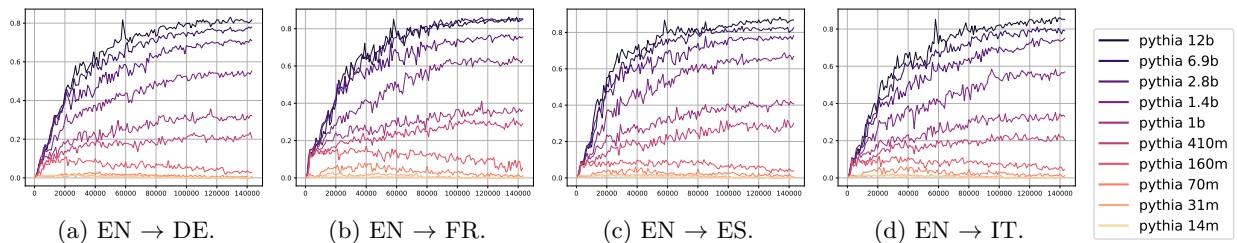

(a) EN → DE.          (b) EN → FR.          (c) EN → ES.          (d) EN → IT.

Figure 7: The Pythia models' ability to perform token translation in the ICL setting developed gradually as the training time and model size scale up.

insufficiently trained. Therefore, the developmental character is fundamental to the model and cannot be bridged by simply scaling up the model, either in size or training time.

## 5.2 ICL Competence for Different Task Configurations Develops in Lock-step

Next, we examine how ICL competence develops across different task configurations. Figure 6 shows the development of ICL competence on WC and LSCG across one of their respective task configurations. Overall, we observe that ICL competence develops in a tightly coupled manner across task configurations, with different setups exhibiting similar trends as training progresses. More interestingly, task performance reaches peaks and valleys at the same training steps, displaying a synchronized pattern over training time. This suggests that, similar to the results on token sampling range, while easier configurations develop faster initially, the performance differences across task configurations eventually stabilize as training progresses. This trend becomes even more apparent when adjusting for the peak performance reached across all checkpoints. *Therefore, the performance's dependence on task configuration (and thus task difficulty) is unlikely to be resolved with further training.*

Apart from visual inspection, we confirm that the development of ICL competence across different task configurations cointegrates using the Johansen's (1991) test. Cointegration implies that two or more time series move together over time, exhibiting simultaneous fluctuations while maintaining a stable long-term relationship. This makes cointegration particularly suitable for studying ICL development in this case, as it identifies true long-term relationships between time series while avoiding spurious correlations driven by general increasing or decreasing trends.

Therefore, our findings in Section 4—that ICL generalization has its limits and depends on token statistics and task configurations—are unlikely to be artifacts of insufficient training. In other words, the true ICL generalization stridently claimed by Olsson et al. (2022) is not likely achievable simply by scaling up with more training or additional model parameters. These characteristics appear to be fundamental to ICL, rather than mere byproducts of training constraints.

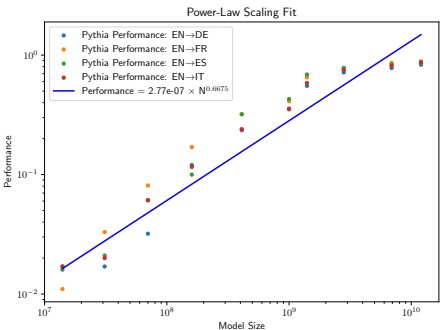
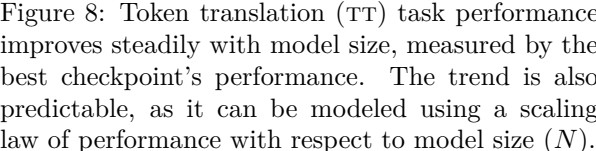
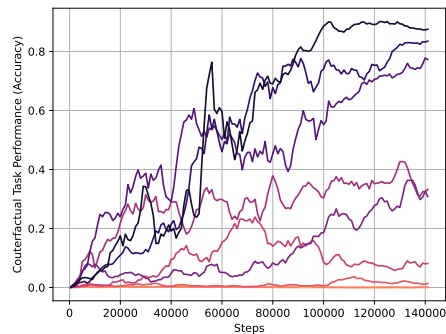

Figure 8: Token translation (TT) task performance improves steadily with model size, measured by the best checkpoint's performance. The trend is also predictable, as it can be modeled using a scaling law of performance with respect to model size ($N$).

Figure 9: The development fluctuates more for the counterfactual (CF) task, which is arguably the most complicated of our proposed tasks. The curve is smoothed with a 5-point running average. Colors show model size, matching Figure 7.

## 5.3 A Scaling Law: the Development of Complex Aspects of ICL is Gradual and Predictable

The two aforementioned findings using these simple, literal pattern-based data corroborate Olsson et al.'s (2022) general observation that there is a period of rapid development of ICL competence near the start of the pre-training process, although we provide a more nuanced interpretation of this development with respect to a finer level of generalization.

Nevertheless, we also observe that when extending the investigation to more complex tasks, such as Token Translation (TT) and Counterfactual (CF)—which require intricate composition and integration of linguistic and factual information—their development happens much later in the pre-training process. This development also appears more gradual and predictable compared to that of literal pattern-based tasks.

Figure 7 shows the development of the token translation task. The rapid development previously observed between steps 512 and 1000 in simpler synthetic tasks is no longer present. Instead, ICL competence continues to develop well into the later stages of the Pythia training process. This development of more complicated ICL tasks can also be described as predictable, since we can fit a *scaling law* in model size (Performance $\sim -2.77 \times 10^{-7} \times N^{0.6675}$, where $N$ is the model size) as shown in Figure 8.

The development fluctuates more for the counterfactual (CF) task, as shown in Figure 9, which is arguably the most complex among our proposed tasks. This is expected, as the task places heavier demands on the model's reasoning capabilities. It requires not only integrating factual knowledge but also overriding memorized or parametrized information in favor of generating token sequences that should, by design, be surprising or counter to the model's prior expectations.

Whether LM abilities develop abruptly and unpredictably, as famously proposed by Wei et al. (2022), or whether the process follows a more predictable pattern (Lu et al., 2024; Schaeffer et al., 2023), remains a topic of heated debate in the interpretability and language modeling community. Our investigation of more complex tasks suggests that ICL competence—an essential component of LM abilities—develops gradually and predictably, following a describable scaling law. The rapid development of ICL competence observed by Olsson et al. (2022), which led to an abrupt phase change, however, is only seen in the easier synthetic tasks in our study. Further insights could be gained by extending our analysis to more fine-grained checkpointing intervals, particularly between steps 512 and 1000. However, to the best of our knowledge, no such publicly available repository exists beyond Pythia. Thus, we leave this for future work.

## 6 ICL Development and the Internal Subspace Allocation

Finally, we present a mechanistic connection between the development of ICL competence and specialization in the residual stream's subspace. Specifically, we find that despite the trajectories of different ICL tasks

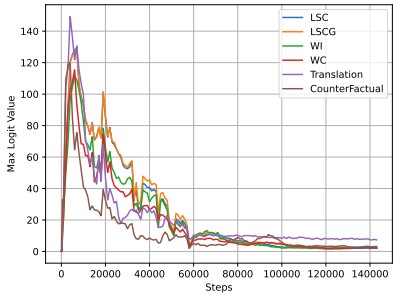 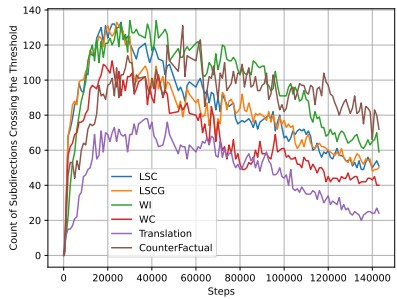 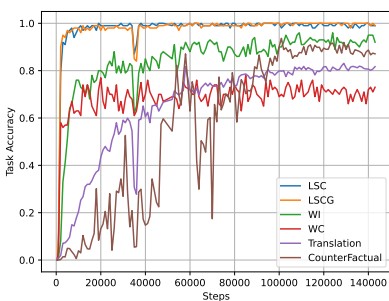

(a) Maximum logit value across singular value directions at each training step. The curve shows a long-tailed shape, with the peak appearing at a very early stage (~1000 steps).

(b) Number of singular value directions with logits crossing the threshold $\tau = 0.2$. We can observe an early rise (~20,000 steps) followed by a gradual decline that remains above zero.

(c) ICL performance development is idiosyncratic across tasks: easier tasks peak early, translation improves gradually, and counterfactual performance fluctuates heavily.

Figure 10: The formation of subspaces in the residual stream follows the same trend (a,b) across all ICL tasks, even though their development differs drastically (c).

varying widely, the trend of the formation of their internal structure remains consistent. Therefore, we hypothesize a connection between the internal mechanism (particularly subspace allocation) and the model's generalization ability, which ultimately forms the foundation of LMs' ICL competence.

## 6.1 Method: Singular Residual Stream Direction Analysis

Here, we present a novel method to study the subspace specialization process through Singular Residual Stream (Unembedding) Direction Analysis (SUDA). First, we perform a singular value decomposition (SVD) of the token unembedding ($W_U$) module[9]: $USV^h = \text{SVD}(W_U)$. Then, we measure how well a single singular vector direction $V_i^h$ can perform the task. Specifically, we compute the logit of the answer token ($t_{\text{ans}}$) using the last layer's residual output: $\ell(t_{\text{ans}}) = V_i^h x_{-1}$.

Thus, for each pre-training Pythia checkpoint, we obtain a logit score for each singular component for every input sample. For `Pythia-12b`[10], the SVD process yields 5120 unique singular directions. As in our prior experiments, we feed the models 1000 samples across all tasks (LSC, LSCG, WI, WC, TT, CF) using their default configurations[11] and take the average. We analyze the checkpoints timestep by timestep. We focus on two key statistics: (1) the maximum logit across all singular directions, and (2) how many "strong" directions exceed a given logit threshold. These two statistics, the max and spread, allow us to characterize how subspaces are utilized.

By observing these two statistics, we are especially interested in characterizing three key characteristics in this SUDA analysis: (1) to what degree individual singular value directions contribute to a task; (2) how many singular value directions the model uses to perform a given task; and (3) how these characteristics develop throughout the pre-training process—are they related to the developmental characteristics we previously identified? We believe this novel method can be applied to a wide range of new tasks within the interpretability domain, and we welcome future work along with this direction.

Table 4: Mean Overlap Matrix between Tasks.

| Task | LSC | LSCG | WI | WC | TT | CF | COUNTRYCAPITAL |
|------|-----|------|-----|-----|-----|-----|----------------|
| LSC | - | 0.632 | 0.597 | 0.593 | 0.303 | 0.224 | 0.207 |
| LSCG | 0.632 | - | 0.608 | 0.578 | 0.305 | 0.215 | 0.206 |
| WI | 0.597 | 0.608 | - | 0.528 | 0.306 | 0.235 | 0.192 |
| WC | 0.593 | 0.578 | 0.528 | - | 0.297 | 0.204 | 0.231 |
| TT | 0.303 | 0.305 | 0.306 | 0.297 | - | 0.144 | 0.181 |
| CF | 0.224 | 0.215 | 0.235 | 0.204 | 0.144 | - | 0.096 |

## 6.2 The Formation of Subspace Specialization Corresponds to ICL Generalization Ability

Figure 10 presents the key findings of our SUDA results. Interestingly, despite the idiosyncratic development process across different ICL tasks, the growth of ICL competence is fairly consistent for all tasks, revealing several key time points that correspond to the development of generalization, as we observed previously.

First, we confirm the underlying premise that certain singular directions can indeed perform the task to a nontrivial extent. Figure 10a shows the logit value achieved by the best-performing direction. We observe that strong directions appear very early in training, with a sharp peak around 1000 steps. This indicates that highly effective sub-directions appear surprisingly early during training, before the model has converged or developed broader capabilities. Interestingly, this early peak coincides with the period when the model develops in-distribution generalization ability most rapidly (§5.1) and when the simplest literal pattern-based task (LSC) reaches near-perfect performance.

On the other hand, when counting the number of high-performing sub-directions—Figure 10b shows how many sub-directions yield logits higher than a threshold ($\tau = 0.2$)—we see a different trend, though one that is also consistent across ICL tasks. Specifically, we observe an early increase in the number of sub-directions crossing the threshold, peaking around 20,000 steps. This is followed by a gradual decline that trend towards stabilization at a non-zero level, suggesting that while the model's reliance on sharply task-relevant directions decreases over time, a persistent subset of effective directions remains active throughout training.

These two trajectories of internal subspace formation closely resemble how ICL generalization ability develops (as we explored in §5). First, recall that in-distribution generalization develops earlier, with the gap between in- and out-of-distribution generalization peaking around 1000 steps and then decreasing in a long-tailed manner. A similar pattern holds for the best-performing subspace's ability to reproduce the full model's performance: the max logit peaks near 1000 steps, followed by a long-tailed decline. We believe this is not a coincidence. It is reasonable to assume that dominant singular value directions capture the most frequent or structurally simple cases—those the model learns to handle correctly early in training. As shown earlier in the paper, ICL ability tends to appear first on common tokens before extending to rarer or more complex ones. In both cases, the metric reflects the performance of the single best-performing component, whether it is a singular value direction or a performance difference across token frequency. The similarity in their trajectories is somewhat surprising but ultimately coherent.

Second, recall that we observe a different curve when switching to Pearson correlation—a more comprehensive measure that captures nuanced levels of in- and out-of-distribution generalization, not just the extremes. We see an analogous trend in Figure 10b when extending the analysis to all singular value directions, not just the maximum. The number of strong singular value directions peaks at around 20,000—the same as in the Pearson correlation result—and is followed by a gradual decrease.

**Are these subspaces tied to ICL?** While these singular value directions are used by the model to perform ICL tasks, our analyses hitherto do not establish any exclusivity. To check whether certain singular value directions resonate with certain tasks, we look into the sharing of subspaces across tasks, defined by

---

[9]We follow the notation and formulation of the residual stream proposed by Elhage et al. (2021).

[10]We obtain the result in this section using the largest and most capable `Pythia-12b` model.

[11]We choose the settings where the task has reasonable complexity while the models can still obtain high performance. The particular settings are LSC: |P|=5, |R|=10; LSC: |P|=5, |R|=10, |G|=2; WI: |S|=5, $i$=1; and WC: |F|=3, |L|=2, |D|=7.

the average intersection over union (IoU) overlap throughout the training steps, defined as $\text{IoU}(A, B) = (|V_A \cap V_B|) / (|V_A \cup V_B|)$, where $A, B$ are two tasks and $V_A, V_B$ are their corresponding sets of singular value directions. As a control, we consider COUNTRYCAPITAL, a factual recall task that asks the model for the capital city of a given country, using the prompt template: `The capital city of <country> is <city>`. We use the underlying factual data from the CF task to generate this task. Unlike the other tasks, it does not require any non-trivial task learning from context.

In Table 4, we show the degree of singular value direction sharing among different tasks. We can see a high degree of sharing between the tasks devoid of any semantics, i.e., LSC, LSCG, WC and WI. Requirement of pretrained knowledge introduces the usage of new subspace, and therefore the singular direction sharing between the non-semantic tasks and token translation (or counterfactual) decreases. For any of these ICL tasks, subspace sharing is lowest with the non-ICL factual recall task. Interestingly, the formation of singular value directions for the COUNTRYCAPITAL task follows a distinctly different trajectory compared to all the ICL tasks (see Figure 13 in Appendix E). We can conclude that: 1) development od ICL is exclusively tied to the formation of certain subspaces in the residual stream, 2) tasks that require pretrained knowledge can be associated with distinctly separate subspaces, and 3) formation (over pre-training) of these two classes of subspaces are distinctly different. We hypothesize that, these subspaces arise from the distinct subspaces of the low-rank attention heads: certain heads specialize in moving information from the context (Olsson et al., 2022) whereas other heads act as movers of pretrained knowledge (Lv et al., 2024; Singh et al., 2024). We leave the training dynamics of the attention head-specific subspaces and their alignment as an important future work.

## 7 Discussion

**ICL's off-distribution generalization ability and its safety implications.** As stated in their discussion section, the ultimate motivation of Olsson et al. (2022) to investigate ICL and its underlying mechanism is to "help us be confident in their [LMs'] safety" (Olsson et al., 2022). They argue that LMs could be unsafe because their abilities should not be seen as static after training ends: the model can acquire new capabilities during inference, without further training, through the mechanism of ICL. They also emphasize the remarkable off-distribution generalization ability demonstrated by ICL, which could lead to surprising and unwanted LM behaviors.

Our surprising finding that models generalize to different degrees of performance based on something as simple as the tokenizer index adds more nuance to this discussion. The results in Section 4 provide strong evidence that ICL has limits when it comes to off-distribution generalization, and that this level of generalization can still be traced back to pre-training. In other words, ICL—like other "traditional" machine learning regimes[12]—does not escape the statistical information it encounters during training, and we reasonably conjecture that there is a theoretical upper bound on LM general ability competence that cannot be surpassed by learning done purely during inference in context.

**ICL's abrupt "emergence" and its safety implications.** The term *emergence*, popularized by Wei et al. (2022), their follow-up work (Ganguli et al., 2022), and even by refutations (Schaeffer et al., 2023; Lu et al., 2024), refers to the abrupt and unpredictable appearance of LM capability during the pre-training process.[13] Wei et al. (2022) observed a "quiet" period during scaling (whether in model size or pre-training data size, typically expressed together as floating point operations) when no meaningful or significant development appears, followed by a sudden abrupt jump in performance after crossing a certain scaling threshold. There is still ongoing debate over whether these observations are justified. Most famously and convincingly, Schaeffer et al. (2023) showed that the development trend is no longer abrupt when switching to a continuous metric such as token log probability or cross-entropy loss. In Section 5.3, we find that various aspects of ICL are not abrupt at all, and can in fact be predicted by a smooth scaling law.

---

[12] Again, we use the term *traditional machine learning regimes* loosely here. Compared to the novel learning methods that operate in context, neural networks themselves are traditional.

[13] There is another commonly used definition of *emergence* in the philosophy of science (Anderson, 1972; Batterman, 2001; O'Connor, 2021; Holtzman et al., 2023), which describes cases where explanatory theories do not translate from the microscopic to the macroscopic scale. We refer interested readers to the relevant discussions.

Our work suggests that ICL *is* a novel property that emerges in LMs during pre-training. While a model's ICL competence appears to increase with scale, our results indicate that ICL becomes detectable as early as after 1000 training steps, suggesting that the development of ICL competence in LMs should be considered predictable. Furthermore, we find that ICL follows a clear scaling law with respect to model size and training time. Some AI safety researchers, concerned about the unpredictable development of AI abilities, have argued that harmful capabilities (e.g., the ability to deceive or manipulate users) could emerge later in training without any early warning signs. Crucially, our findings challenge this view, showing that ICL develops in a measurable and transparent manner.

## 8 Conclusion

In this paper, we present a thorough investigation into two fundamental research questions of ICL: what it is and how it develops during pre-training. We find that ICL is neither a mere illusion of superficial memorization nor the emergence of some full symbolic algorithm. Instead, it exhibits a peculiar mixture of generalization and dependence on token statistics. This mixture is likely an inherent feature of ICL and LMs, as its development stabilizes after an initial phase of rapid growth during pre-training.

We also provide a more nuanced and accurate characterization on the development of ICL, tracing it back to the formation of internal mechanisms in LMs. In particular, we observe that different aspects of ICL competence develop at different rates, generally following the trend that "easier" aspects arise earlier and appear more rapid, while more complex aspects take longer to develop, presenting as gradual and predictable. These development are ultimately linked to the formation of internal mechanisms, which can be analyzed through the allocation of subspaces in the residual stream. Therefore, our analysis goes beyond superficially probing model performance; instead, we propose a viable approach to bridging training dynamics analysis and mechanistic interpretability.

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

Table 5: Sampled Tokens (Part 1).

English: apple, banana, pear, apricot, plum, cherry, grape, kiwi, fig, orange, mandarin, strawberry, raspberry, carrot, onion, garlic, pepper, cucumber, tomato, lettuce, cabbage, mushroom, bean, pea, cat, dog, horse, cow, pig, sheep, bull, rabbit, tiger, giraffe, monkey, bird, fish, whale, frog, snake, car, bus, train, ship, boat, bicycle, motorcycle, truck, table, chair, bed, book, pen, pencil, paper, letter, envelope, clock, bank, store, beach, bottle, cup, plate, glass, fork, knife, pot, lake, sea, rock, sand, flower, leaf, grass, arm, hand, leg, ear, nose, mouth, tooth, shirt, shoe, love, hope, peace, war, time, month, week, zero, one, two, three, four, six, seven, eight, nine, radio, computer, printer, keyboard, red, blue, yellow, black, white, brown, gray, pink, question, january, april, may, june, july, september, october, november, tennis, basketball, golf, bread, water, milk, cheese, butter, egg, meat, chicken, juice, wine, chocolate, ring, stone, money, banknote, stamp, album, music, desk, folder, file, form, nurse, teacher, government, village, house, apartment, work, job, company, factory, restaurant, bar, park, zoo, road, highway, harbor, rain, snow, wind, storm, cloud, sky, sun, moon, galaxy, universe, map, flag, language, word, sentence, magazine, cinema, photo, image, video, web, internet, mail, message, game, toy, ball, gift, festival, vacation, wedding, family, friend, enemy, prince, princess, palace.

German: Apfel, Banane, Birne, Aprikose, Pflaume, Kirsche, Traube, Kiwi, Feige, Orange, Mandarine, Erdbeere, Himbeere, Karotte, Zwiebel, Knoblauch, Paprika, Gurke, Tomate, Salat, Kohl, Pilz, Bohne, Erbse, Katze, Hund, Pferd, Kuh, Schwein, Schaf, Stier, Kaninchen, Tiger, Giraffe, Affe, Vogel, Fisch, Wal, Frosch, Schlange, Auto, Bus, Zug, Schiff, Boot, Fahrrad, Motorrad, Laster, Tisch, Stuhl, Bett, Buch, Stift, Bleistift, Papier, Brief, Umschlag, Uhr, Bank, Laden, Strand, Flasche, Tasse, Teller, Glas, Gabel, Messer, Topf, See, Meer, Fels, Sand, Blume, Blatt, Gras, Arm, Hand, Bein, Ohr, Nase, Mund, Zahn, Hemd, Schuh, Liebe, Hoffnung, Frieden, Krieg, Zeit, Monat, Woche, Null, Eins, Zwei, Drei, Vier, Sechs, Sieben, Acht, Neun, Radio, Computer, Drucker, Tastatur, Rot, Blau, Gelb, Schwarz, Weiss, Braun, Grau, Rosa, Frage, Januar, April, Mai, Juni, Juli, September, Oktober, November, Tennis, Basketball, Golf, Brot, Wasser, Milch, Kaese, Butter, Ei, Fleisch, Huhn, Saft, Wein, Schokolade, Ring, Stein, Geld, Geldschein, Briefmarke, Album, Musik, Schreibtisch, Ordner, Datei, Formular, Pfleger, Lehrer, Regierung, Dorf, Haus, Wohnung, Arbeit, Job, Firma, Fabrik, Restaurant, Bar, Park, Zoo, Strasse, Autobahn, Hafen, Regen, Schnee, Wind, Sturm, Wolke, Himmel, Sonne, Mond, Galaxie, Universum, Karte, Fahne, Sprache, Wort, Satz, Zeitschrift, Kino, Foto, Bild, Video, Web, Internet, Post, Nachricht, Spiel, Spielzeug, Ball, Geschenk, Festival, Urlaub, Hochzeit, Familie, Freund, Feind, Prinz, Prinzessin, Palast.

# A   Details of the Token Translation Task

For the token translation task, we sample 200 common English nouns and translate them into German, French, Spanish, or Italian using ChatGPT[14]. We first ask ChatGPT to generate 500 tokens using the following prompt for sampling:

```
Give me 500 simple English nouns and their translation in German[15].  Words
should be really simple for a beginner for both languages.  Give me two
Python lists.
```

Then, we manually examine the results generated by ChatGPT and narrow the list down to 200. Specifically, we prefer those that can be tokenised into a single token. The selected tokens are shown in Table 5 and 6.

---

[14]https://chatgpt.com/
[15]We replace the language "German" with the specific lanugage in each cases.

Table 6: Sampled Tokens (Part 2).

French: pomme, banane, poire, abricot, prune, cerise, raisin, kiwi, figue, orange, mandarine, fraise, framboise, carotte, oignon, ail, poivron, concombre, tomate, laitue, chou, champignon, haricot, pois, chat, chien, cheval, vache, cochon, mouton, taureau, lapin, tigre, girafe, singe, oiseau, poisson, baleine, grenouille, serpent, voiture, bus, train, navire, bateau, bicyclette, moto, camion, table, chaise, lit, livre, stylo, crayon, papier, lettre, enveloppe, horloge, banque, magasin, plage, bouteille, tasse, assiette, verre, fourchette, couteau, marmite, lac, mer, roche, sable, fleur, feuille, herbe, bras, main, jambe, oreille, nez, bouche, dent, chemise, chaussure, amour, espoir, paix, guerre, temps, mois, semaine, zero, un, deux, trois, quatre, six, sept, huit, neuf, radio, ordinateur, imprimante, clavier, rouge, bleu, jaune, noir, blanc, marron, gris, rose, question, janvier, avril, mai, juin, juillet, septembre, octobre, novembre, tennis, basketball, golf, pain, eau, lait, fromage, beurre, oeuf, viande, poulet, jus, vin, chocolat, bague, pierre, argent, billet, timbre, album, musique, bureau, dossier, fichier, formulaire, infirmier, enseignant, gouvernement, village, maison, appartement, travail, emploi, entreprise, usine, restaurant, bar, parc, zoo, route, autoroute, port, pluie, neige, vent, orage, nuage, ciel, soleil, lune, galaxie, univers, carte, drapeau, langue, mot, phrase, magazine, cinema, photo, image, video, web, internet, courrier, message, jeu, jouet, balle, cadeau, festival, vacances, mariage, famille, ami, ennemi, prince, princesse, palais

Italian: mela, banana, pera, albicocca, prugna, ciliegia, uva, kiwi, fico, arancia, mandarino, fragola, lampone, carota, cipolla, aglio, peperone, cetriolo, pomodoro, lattuga, cavolo, fungo, fagiolo, pisello, gatto, cane, cavallo, mucca, maiale, pecora, toro, coniglio, tigre, giraffa, scimmia, uccello, pesce, balena, rana, serpente, auto, bus, treno, nave, barca, bicicletta, moto, camion, tavolo, sedia, letto, libro, penna, matita, carta, lettera, busta, orologio, banca, negozio, spiaggia, bottiglia, tazza, piatto, bicchiere, forchetta, coltello, pentola, lago, mare, roccia, sabbia, fiore, foglia, erba, braccio, mano, gamba, orecchio, naso, bocca, dente, camicia, scarpa, amore, speranza, pace, guerra, tempo, mese, settimana, zero, uno, due, tre, quattro, sei, sette, otto, nove, radio, computer, stampante, tastiera, rosso, blu, giallo, nero, bianco, marrone, grigio, rosa, domanda, gennaio, aprile, maggio, giugno, luglio, settembre, ottobre, novembre, tennis, basket, golf, pane, acqua, latte, formaggio, burro, uovo, carne, pollo, succo, vino, cioccolato, anello, pietra, denaro, banconota, francobollo, album, musica, scrivania, cartella, file, formulario, infermiere, insegnante, governo, villaggio, casa, appartamento, lavoro, mestiere, azienda, fabbrica, ristorante, bar, parco, zoo, strada, autostrada, porto, pioggia, neve, vento, tempesta, nuvola, cielo, sole, luna, galassia, universo, mappa, bandiera, lingua, parola, frase, rivista, cinema, foto, immagine, video, web, internet, posta, messaggio, gioco, giocattolo, palla, regalo, festival, vacanza, matrimonio, famiglia, amico, nemico, principe, principessa, palazzo.

Spanish: manzana, banana, pera, albaricoque, ciruela, cereza, uva, kiwi, higo, naranja, mandarina, fresa, frambuesa, zanahoria, cebolla, ajo, pimiento, pepino, tomate, lechuga, col, hongo, frijol, guisante, gato, perro, caballo, vaca, cerdo, oveja, toro, conejo, tigre, jirafa, mono, ave, pez, ballena, rana, serpiente, coche, bus, tren, barco, bote, bicicleta, moto, camioneta, mesa, silla, cama, libro, pluma, lapicero, papel, carta, sobre, reloj, banco, tienda, playa, botella, taza, plato, vaso, tenedor, cuchillo, olla, lago, mar, roca, arena, flor, hoja, hierba, brazo, mano, pierna, oreja, nariz, boca, diente, camisa, zapato, amor, esperanza, paz, guerra, tiempo, mes, semana, cero, uno, dos, tres, cuatro, seis, siete, ocho, nueve, radio, ordenador, impresora, teclado, rojo, azul, amarillo, negro, blanco, marron, gris, rosa, pregunta, enero, abril, mayo, junio, julio, septiembre, octubre, noviembre, tenis, baloncesto, golf, pan, agua, leche, queso, mantequilla, huevo, carne, pollo, jugo, vino, chocolate, anillo, piedra, dinero, billete, sello, album, musica, escritorio, carpeta, archivo, formulario, enfermero, maestro, gobierno, pueblo, casa, apartamento, trabajo, empleo, empresa, fabrica, restaurante, bar, parque, zoo, camino, autopista, puerto, lluvia, nieve, viento, tormenta, nube, cielo, sol, luna, galaxia, universo, mapa, bandera, lengua, palabra, frase, revista, cine, foto, imagen, video, web, internet, correo, mensaje, juego, juguete, pelota, regalo, festival, vacaciones, boda, familia, amigo, enemigo, principe, princesa, palacio

# B  Supplementary Experiment Results

In this section, we present a more comprehensive collection of results. In particular, Table 7 shows the results of LSC, Table 8 shows LSCG, Table 9 shows WI, and Table 10 also shows WC. These complementary results provide a more complete view of the models' performance, supporting the same findings elaborated in Table 2. The complete list of results, comprising all combinations of task configuration settings, is released together with our code.

Table 7: Models' Performance on the LSC Task across Different Task Configurations.

| Setting | | | Pythia Models Accuracy (%) | | | | | | | | | | LLaMA Models Accuracy (%) | | | |
|---|---|---|---|---|---|---|---|---|---|---|---|---|---|---|---|---|
| $|P|$ | $|R|$ | 14M | 31M | 70M | 160M | 410M | 1B | 1.4B | 2.8B | 6.9B | 12B | 3.2-1B | 3.2-3B | 3.1-8B | 3.1-8B-Inst |
| 1 | 1 | 0.6 | 1.1 | 1.4 | 4.1 | 38.1 | 43.1 | 48.2 | 52.1 | 51.4 | 62.0 | 69.6 | 75.3 | 82.0 | 63.2 |
| 1 | 2 | 0.7 | 0.9 | 1.9 | 5.4 | 51.2 | 53.6 | 57.6 | 60.9 | 64.5 | 73.1 | 74.7 | 77.3 | 84.4 | 60.0 |
| 1 | 5 | 1.0 | 0.9 | 0.9 | 7.0 | 69.4 | 65.9 | 67.5 | 61.7 | 76.7 | 82.5 | 71.7 | 78.7 | 86.1 | 45.6 |
| 1 | 10 | 0.6 | 0.6 | 2.1 | 7.8 | 69.4 | 73.7 | 61.8 | 53.6 | 80.4 | 82.2 | 71.3 | 77.5 | 92.0 | 39.6 |
| 1 | 20 | 1.4 | 1.4 | 1.2 | 7.2 | 67.8 | 71.7 | 56.4 | 50.3 | 72.4 | 83.0 | 67.2 | 74.5 | 89.9 | 30.9 |
| 2 | 1 | 1.4 | 2.9 | 4.9 | 18.9 | 73.2 | 85.5 | 78.6 | 78.7 | 84.1 | 90.8 | 92.1 | 94.5 | 97.9 | 93.9 |
| 2 | 2 | 1.1 | 1.2 | 3.8 | 19.3 | 81.6 | 89.6 | 83.4 | 82.1 | 88.0 | 93.3 | 95.1 | 96.3 | 98.4 | 95.9 |
| 2 | 5 | 1.1 | 1.3 | 2.8 | 20.9 | 91.3 | 92.5 | 90.2 | 86.8 | 93.5 | 97.2 | 96.9 | 97.8 | 99.4 | 96.5 |
| 2 | 10 | 1.1 | 0.9 | 3.7 | 21.8 | 91.3 | 95.5 | 89.1 | 88.4 | 95.9 | 97.6 | 97.0 | 98.0 | 99.7 | 96.9 |
| 2 | 20 | 1.9 | 1.5 | 2.4 | 25.1 | 93.4 | 95.6 | 88.0 | 93.9 | 96.5 | 98.9 | 98.0 | 98.8 | 99.7 | 97.2 |
| 5 | 1 | 1.4 | 2.6 | 5.8 | 37.5 | 91.9 | 92.1 | 88.4 | 91.9 | 95.1 | 97.6 | 97.6 | 99.4 | 99.8 | 99.9 |
| 5 | 2 | 0.9 | 1.7 | 5.2 | 40.0 | 92.1 | 93.4 | 91.5 | 95.8 | 97.4 | 97.8 | 98.9 | 99.5 | 100.0 | 99.8 |
| 5 | 5 | 2.1 | 1.7 | 6.5 | 45.4 | 96.3 | 95.8 | 93.0 | 95.1 | 97.7 | 98.4 | 99.7 | 100.0 | 100.0 | 99.9 |
| 5 | 10 | 3.2 | 2.6 | 5.9 | 48.5 | 97.5 | 98.0 | 95.7 | 98.0 | 98.6 | 99.6 | 98.9 | 99.9 | 100.0 | 99.9 |
| 5 | 20 | 2.4 | 0.6 | 5.6 | 51.5 | 98.9 | 98.4 | 95.4 | 98.7 | 98.8 | 99.0 | 99.4 | 100.0 | 100.0 | 99.8 |
| 10 | 1 | 2.3 | 3.2 | 6.2 | 57.4 | 97.1 | 95.6 | 94.4 | 98.0 | 98.3 | 99.4 | 98.7 | 99.8 | 99.9 | 99.5 |
| 10 | 2 | 2.6 | 2.2 | 6.1 | 59.7 | 96.9 | 96.1 | 95.5 | 98.6 | 98.4 | 99.9 | 99.5 | 99.9 | 100.0 | 99.9 |
| 10 | 5 | 2.0 | 1.8 | 7.3 | 58.4 | 97.8 | 97.2 | 96.5 | 99.1 | 99.5 | 99.9 | 99.9 | 100.0 | 99.9 | 99.9 |
| 10 | 10 | 3.0 | 1.6 | 4.8 | 51.8 | 99.3 | 98.6 | 98.7 | 99.7 | 99.8 | 99.9 | 100.0 | 100.0 | 100.0 | 100.0 |
| 10 | 20 | 3.0 | 3.2 | 6.4 | 60.9 | 99.2 | 99.6 | 97.3 | 99.7 | 99.8 | 100.0 | 99.9 | 100.0 | 100.0 | 99.9 |
| 20 | 1 | 2.3 | 7.4 | 3.7 | 58.8 | 99.1 | 99.5 | 98.6 | 98.8 | 99.1 | 99.9 | 95.1 | 99.1 | 99.9 | 98.8 |
| 20 | 2 | 2.6 | 5.3 | 4.2 | 62.9 | 98.4 | 99.4 | 97.8 | 99.2 | 99.1 | 99.8 | 97.4 | 99.8 | 99.7 | 99.8 |
| 20 | 5 | 2.1 | 6.9 | 3.6 | 63.8 | 99.4 | 99.9 | 99.3 | 99.9 | 99.7 | 99.9 | 98.9 | 100.0 | 100.0 | 99.9 |
| 20 | 10 | 2.7 | 6.4 | 3.4 | 60.2 | 99.4 | 99.5 | 99.4 | 99.7 | 99.8 | 100.0 | 100.0 | 99.9 | 100.0 | 100.0 |
| 20 | 20 | 2.1 | 7.2 | 2.4 | 60.0 | 99.2 | 99.7 | 99.7 | 99.8 | 99.8 | 100.0 | 99.9 | 100.0 | 100.0 | 100.0 |

Table 8: Models' Performance on the LSCG Task across Different Task Configurations.

| Setting | | | Pythia Models Accuracy (%) | | | | | | | | | | LLaMA Models Accuracy (%) | | | |
|---|---|---|---|---|---|---|---|---|---|---|---|---|---|---|---|---|---|
| \|P\| | \|R\| | \|G\| | 14M | 31M | 70M | 160M | 410M | 1B | 1.4B | 2.8B | 6.9B | 12B | 3.2-1B | 3.2-3B | 3.1-8B | 3.1-8B-Inst |
| 5 | 5 | 0 | 1.8 | 1.6 | 5.5 | 53.1 | 96.1 | 96.9 | 94.3 | 98.5 | 98.4 | 99.5 | 99.8 | 100.0 | 100.0 | 100.0 |
| 5 | 5 | 5 | 1.0 | 1.1 | 0.5 | 14.7 | 58.4 | 63.1 | 68.6 | 79.1 | 73.8 | 81.1 | 89.5 | 94.8 | 96.5 | 87.9 |
| 5 | 5 | 10 | 0.7 | 0.4 | 0.6 | 7.9 | 39.7 | 51.2 | 49.1 | 61.2 | 58.5 | 68.0 | 73.9 | 83.9 | 89.5 | 56.5 |
| 5 | 10 | 0 | 2.4 | 0.9 | 5.9 | 51.3 | 97.9 | 97.5 | 97.0 | 99.2 | 98.4 | 99.5 | 99.7 | 100.0 | 100.0 | 100.0 |
| 5 | 10 | 5 | 1.1 | 1.0 | 0.9 | 10.8 | 52.7 | 60.5 | 66.2 | 79.1 | 74.1 | 80.3 | 91.1 | 94.1 | 96.8 | 87.7 |
| 5 | 10 | 10 | 0.8 | 0.6 | 0.4 | 6.4 | 39.3 | 47.9 | 55.5 | 59.1 | 55.3 | 66.1 | 74.5 | 87.2 | 90.8 | 53.9 |
| 5 | 20 | 0 | 2.1 | 1.7 | 4.6 | 47.6 | 98.6 | 98.8 | 97.4 | 99.7 | 99.3 | 99.7 | 99.7 | 100.0 | 100.0 | 100.0 |
| 5 | 20 | 5 | 0.9 | 0.9 | 0.3 | 9.4 | 52.0 | 54.0 | 63.3 | 77.6 | 70.2 | 82.6 | 92.4 | 95.8 | 96.3 | 86.5 |
| 5 | 20 | 10 | 1.4 | 0.5 | 0.3 | 3.2 | 25.9 | 42.7 | 40.1 | 56.5 | 47.6 | 65.5 | 82.4 | 90.9 | 92.5 | 59.3 |
| 10 | 5 | 0 | 2.0 | 2.2 | 5.8 | 57.7 | 97.2 | 98.6 | 98.3 | 99.5 | 99.5 | 99.9 | 99.9 | 100.0 | 100.0 | 100.0 |
| 10 | 5 | 5 | 1.5 | 0.9 | 0.4 | 17.9 | 63.9 | 73.6 | 76.3 | 88.6 | 75.6 | 84.1 | 94.6 | 97.2 | 97.2 | 92.0 |
| 10 | 5 | 10 | 0.7 | 0.6 | 0.7 | 10.4 | 47.9 | 57.0 | 68.2 | 73.1 | 56.4 | 66.6 | 88.9 | 92.3 | 93.0 | 71.8 |
| 10 | 10 | 0 | 2.4 | 2.6 | 4.0 | 55.4 | 98.4 | 99.3 | 99.0 | 99.6 | 99.8 | 99.9 | 100.0 | 100.0 | 100.0 | 100.0 |
| 10 | 10 | 5 | 1.5 | 0.9 | 0.6 | 17.9 | 64.9 | 71.9 | 79.2 | 86.5 | 74.1 | 84.4 | 96.6 | 97.8 | 97.3 | 93.2 |
| 10 | 10 | 10 | 1.1 | 0.9 | 0.2 | 8.7 | 45.2 | 55.6 | 61.9 | 76.1 | 58.9 | 70.7 | 88.7 | 93.7 | 93.7 | 73.0 |
| 10 | 20 | 0 | 1.5 | 2.9 | 4.4 | 64.2 | 98.9 | 99.3 | 98.2 | 99.9 | 99.8 | 100.0 | 100.0 | 99.9 | 100.0 | 100.0 |
| 10 | 20 | 5 | 0.6 | 0.8 | 0.2 | 11.7 | 55.6 | 70.7 | 74.5 | 87.5 | 66.8 | 86.7 | 97.8 | 98.2 | 98.3 | 95.1 |
| 10 | 20 | 10 | 0.6 | 0.5 | 0.1 | 5.7 | 30.1 | 51.5 | 56.8 | 71.4 | 49.1 | 64.2 | 90.1 | 96.0 | 94.0 | 80.4 |
| 20 | 5 | 0 | 2.1 | 6.5 | 3.7 | 64.1 | 99.3 | 99.4 | 98.9 | 99.5 | 99.8 | 100.0 | 98.3 | 100.0 | 99.9 | 100.0 |
| 20 | 5 | 5 | 1.6 | 1.0 | 0.4 | 15.8 | 68.8 | 82.8 | 84.5 | 90.0 | 75.3 | 80.2 | 94.5 | 98.2 | 95.8 | 92.3 |
| 20 | 5 | 10 | 1.1 | 0.4 | 0.2 | 8.6 | 44.7 | 74.5 | 70.6 | 75.9 | 48.4 | 64.1 | 91.6 | 95.9 | 93.0 | 79.7 |
| 20 | 10 | 0 | 2.1 | 7.4 | 2.8 | 66.4 | 99.6 | 99.2 | 99.7 | 99.8 | 100.0 | 100.0 | 99.4 | 100.0 | 100.0 | 100.0 |
| 20 | 10 | 5 | 1.7 | 0.4 | 0.0 | 12.3 | 60.2 | 83.1 | 84.8 | 86.9 | 68.7 | 81.2 | 96.5 | 98.6 | 97.4 | 95.6 |
| 20 | 10 | 10 | 0.8 | 0.8 | 0.0 | 6.4 | 45.7 | 70.5 | 69.0 | 75.6 | 47.5 | 64.0 | 90.7 | 96.2 | 93.1 | 85.1 |
| 20 | 20 | 0 | 1.8 | 5.9 | 1.9 | 55.7 | 99.6 | 99.6 | 99.7 | 99.9 | 99.9 | 100.0 | 100.0 | 100.0 | 100.0 | 100.0 |
| 20 | 20 | 5 | 1.2 | 1.0 | 0.1 | 8.9 | 61.2 | 79.1 | 78.6 | 87.6 | 65.7 | 80.2 | 95.6 | 97.9 | 96.6 | 95.8 |
| 20 | 20 | 10 | 0.4 | 0.2 | 0.1 | 5.6 | 43.2 | 67.9 | 70.3 | 73.4 | 50.4 | 64.8 | 92.7 | 96.3 | 95.5 | 85.3 |

Table 9: Models' Performance on the WI Task across Different Task Configurations.

| Setting | | Pythia Models Accuracy (%) | | | | | | | | | | LLaMA Models Accuracy (%) | | | |
|---|---|---|---|---|---|---|---|---|---|---|---|---|---|---|---|
| $|P|$ | $i$ | 14M | 31M | 70M | 160M | 410M | 1B | 1.4B | 2.8B | 6.9B | 12B | 3.2-1B | 3.2-3B | 3.1-8B | 3.1-8B-Inst |
| 5 | 0 | 0.0 | 0.0 | 0.1 | 19.5 | 50.2 | 62.7 | 81.0 | 78.8 | 63.9 | 78.3 | 25.1 | 48.9 | 79.7 | 85.1 |
| 5 | 1 | 0.0 | 0.0 | 0.0 | 1.9 | 6.9 | 7.3 | 16.7 | 16.8 | 20.7 | 23.8 | 8.3 | 31.7 | 40.0 | 30.6 |
| 5 | 2 | 0.0 | 0.3 | 0.0 | 2.2 | 8.4 | 11.7 | 15.5 | 14.2 | 22.6 | 25.2 | 12.7 | 30.9 | 34.9 | 27.6 |
| 5 | 3 | 0.0 | 0.0 | 0.0 | 3.8 | 16.5 | 21.5 | 24.1 | 21.5 | 24.6 | 26.9 | 22.1 | 23.2 | 30.6 | 20.6 |
| 5 | 4 | 0.0 | 0.1 | 0.1 | 17.4 | 62.7 | 78.1 | 77.6 | 89.2 | 82.5 | 88.9 | 95.1 | 97.8 | 98.9 | 98.5 |
| 10 | 0 | 0.0 | 0.0 | 0.0 | 4.1 | 28.3 | 31.3 | 37.5 | 33.8 | 24.8 | 53.5 | 6.6 | 41.4 | 54.7 | 66.1 |
| 10 | 1 | 0.0 | 0.0 | 0.0 | 0.0 | 2.9 | 3.2 | 4.1 | 4.4 | 6.7 | 12.7 | 2.7 | 12.7 | 13.1 | 12.6 |
| 10 | 2 | 0.0 | 0.0 | 0.0 | 0.0 | 2.4 | 3.0 | 4.0 | 5.3 | 6.8 | 11.7 | 3.1 | 11.8 | 12.3 | 14.9 |
| 10 | 3 | 0.0 | 0.0 | 0.0 | 0.4 | 1.7 | 2.9 | 4.7 | 5.3 | 8.4 | 11.7 | 4.0 | 17.6 | 13.6 | 15.5 |
| 10 | 4 | 0.0 | 0.0 | 0.0 | 0.7 | 1.6 | 3.5 | 6.2 | 6.5 | 7.1 | 10.3 | 6.8 | 21.8 | 17.9 | 18.9 |
| 10 | 5 | 0.0 | 0.0 | 0.0 | 0.6 | 1.6 | 3.4 | 9.0 | 8.5 | 11.1 | 10.8 | 8.2 | 21.3 | 24.4 | 21.7 |
| 10 | 6 | 0.0 | 0.0 | 0.0 | 1.8 | 2.6 | 4.9 | 10.0 | 10.7 | 12.4 | 13.7 | 12.0 | 23.3 | 25.6 | 22.7 |
| 10 | 7 | 0.0 | 0.0 | 0.0 | 2.0 | 4.5 | 7.6 | 11.0 | 12.6 | 17.7 | 17.1 | 14.9 | 27.7 | 27.0 | 25.5 |
| 10 | 8 | 0.0 | 0.0 | 0.3 | 5.2 | 13.4 | 19.4 | 18.2 | 21.5 | 22.5 | 23.5 | 21.6 | 26.4 | 23.7 | 14.0 |
| 10 | 9 | 0.0 | 0.0 | 0.2 | 15.3 | 59.4 | 73.5 | 71.1 | 85.0 | 84.7 | 89.4 | 95.7 | 99.4 | 99.5 | 99.1 |
| 15 | 0 | 0.0 | 0.0 | 0.2 | 1.3 | 7.7 | 9.5 | 7.8 | 10.1 | 5.3 | 27.9 | 2.5 | 41.9 | 54.3 | 56.7 |
| 15 | 1 | 0.0 | 0.0 | 0.0 | 0.0 | 0.6 | 1.2 | 0.9 | 2.1 | 2.3 | 4.3 | 1.4 | 8.5 | 10.4 | 10.6 |
| 15 | 2 | 0.0 | 0.0 | 0.0 | 0.0 | 0.6 | 1.0 | 1.0 | 1.3 | 2.9 | 3.3 | 2.4 | 7.5 | 8.1 | 6.7 |
| 15 | 3 | 0.0 | 0.0 | 0.0 | 0.1 | 1.0 | 1.0 | 0.7 | 1.1 | 3.5 | 4.6 | 2.3 | 8.5 | 8.4 | 7.1 |
| 15 | 4 | 0.0 | 0.0 | 0.0 | 0.1 | 0.9 | 1.1 | 1.2 | 1.6 | 4.4 | 4.5 | 3.8 | 9.1 | 8.0 | 7.8 |
| 15 | 5 | 0.0 | 0.0 | 0.1 | 0.0 | 0.5 | 1.3 | 2.3 | 2.3 | 5.1 | 5.7 | 4.9 | 10.6 | 9.5 | 8.4 |
| 15 | 6 | 0.0 | 0.0 | 0.0 | 0.0 | 1.2 | 1.8 | 2.9 | 3.0 | 4.6 | 6.7 | 4.1 | 11.4 | 12.4 | 11.8 |
| 15 | 7 | 0.0 | 0.0 | 0.0 | 0.1 | 1.4 | 2.1 | 3.7 | 3.3 | 6.8 | 8.6 | 5.5 | 11.9 | 14.7 | 10.1 |
| 15 | 8 | 0.0 | 0.0 | 0.0 | 0.4 | 1.8 | 2.4 | 4.4 | 4.0 | 7.5 | 8.1 | 6.3 | 14.7 | 14.7 | 11.9 |
| 15 | 9 | 0.0 | 0.0 | 0.1 | 0.4 | 2.1 | 3.6 | 7.0 | 4.7 | 9.8 | 8.0 | 7.0 | 13.6 | 18.7 | 15.0 |
| 15 | 10 | 0.0 | 0.0 | 0.0 | 0.3 | 1.6 | 4.5 | 8.4 | 7.4 | 10.0 | 9.0 | 9.7 | 15.8 | 21.1 | 15.1 |
| 15 | 11 | 0.0 | 0.0 | 0.0 | 1.2 | 2.2 | 5.2 | 9.5 | 10.0 | 12.5 | 10.6 | 12.4 | 18.1 | 23.6 | 17.3 |
| 15 | 12 | 0.0 | 0.0 | 0.0 | 1.4 | 2.2 | 8.7 | 8.5 | 11.1 | 14.6 | 14.6 | 20.3 | 24.6 | 26.8 | 20.6 |
| 15 | 13 | 0.0 | 0.0 | 0.0 | 4.0 | 6.9 | 17.7 | 14.4 | 19.5 | 20.4 | 21.0 | 24.3 | 24.5 | 21.5 | 13.3 |
| 15 | 14 | 0.0 | 0.0 | 0.1 | 16.2 | 62.9 | 72.3 | 66.0 | 86.3 | 87.9 | 91.1 | 96.8 | 99.6 | 99.7 | 98.7 |
| 20 | 0 | 0.0 | 0.0 | 0.0 | 0.8 | 4.3 | 3.8 | 3.4 | 2.8 | 3.4 | 14.6 | 1.5 | 37.3 | 48.3 | 53.3 |
| 20 | 1 | 0.0 | 0.0 | 0.0 | 0.0 | 0.5 | 0.4 | 0.4 | 0.7 | 1.5 | 3.0 | 1.2 | 6.4 | 10.5 | 8.5 |
| 20 | 2 | 0.0 | 0.0 | 0.0 | 0.0 | 0.7 | 0.3 | 0.9 | 0.6 | 2.4 | 2.4 | 0.6 | 4.3 | 6.9 | 5.0 |
| 20 | 3 | 0.0 | 0.0 | 0.0 | 0.0 | 0.7 | 0.2 | 1.0 | 0.7 | 2.3 | 3.4 | 1.7 | 5.2 | 5.0 | 3.6 |
| 20 | 4 | 0.0 | 0.0 | 0.0 | 0.1 | 0.6 | 0.7 | 0.8 | 1.1 | 3.0 | 3.0 | 2.0 | 4.6 | 5.6 | 4.5 |
| 20 | 5 | 0.0 | 0.0 | 0.0 | 0.0 | 0.3 | 0.3 | 0.9 | 0.9 | 1.8 | 3.0 | 2.5 | 4.1 | 5.9 | 3.6 |
| 20 | 6 | 0.0 | 0.0 | 0.0 | 0.0 | 0.2 | 0.8 | 0.7 | 1.0 | 2.6 | 1.9 | 2.4 | 6.7 | 5.8 | 4.0 |
| 20 | 7 | 0.0 | 0.0 | 0.0 | 0.0 | 0.1 | 0.6 | 0.8 | 1.0 | 2.1 | 2.5 | 3.2 | 6.9 | 6.2 | 3.9 |
| 20 | 8 | 0.0 | 0.0 | 0.0 | 0.0 | 0.5 | 0.9 | 1.5 | 1.5 | 3.4 | 3.0 | 2.2 | 7.6 | 8.0 | 6.2 |
| 20 | 9 | 0.0 | 0.0 | 0.0 | 0.0 | 0.4 | 0.9 | 1.2 | 1.3 | 3.8 | 3.8 | 3.7 | 7.9 | 8.4 | 6.9 |
| 20 | 10 | 0.0 | 0.0 | 0.0 | 0.1 | 1.0 | 1.1 | 2.4 | 2.8 | 5.7 | 5.0 | 4.0 | 9.1 | 9.4 | 6.9 |
| 20 | 11 | 0.0 | 0.0 | 0.1 | 0.0 | 1.3 | 2.1 | 2.6 | 2.8 | 6.2 | 5.2 | 4.6 | 9.0 | 11.3 | 7.0 |
| 20 | 12 | 0.0 | 0.0 | 0.0 | 0.4 | 0.9 | 2.0 | 3.7 | 3.4 | 6.5 | 6.4 | 4.9 | 10.8 | 12.4 | 9.6 |
| 20 | 13 | 0.0 | 0.0 | 0.0 | 0.3 | 0.8 | 2.9 | 4.4 | 4.5 | 7.5 | 6.8 | 6.5 | 12.5 | 13.5 | 10.5 |
| 20 | 14 | 0.0 | 0.0 | 0.0 | 0.2 | 1.0 | 2.9 | 4.5 | 5.3 | 9.2 | 6.4 | 8.2 | 14.9 | 14.5 | 12.5 |
| 20 | 15 | 0.0 | 0.0 | 0.0 | 0.4 | 0.8 | 3.5 | 6.7 | 5.4 | 12.3 | 8.4 | 8.4 | 15.7 | 19.5 | 13.6 |
| 20 | 16 | 0.0 | 0.0 | 0.0 | 0.6 | 1.3 | 5.2 | 5.9 | 7.8 | 12.5 | 8.5 | 16.8 | 18.0 | 20.9 | 17.2 |
| 20 | 17 | 0.0 | 0.0 | 0.1 | 2.0 | 3.7 | 9.4 | 9.3 | 12.4 | 15.7 | 14.5 | 20.4 | 25.8 | 25.9 | 18.9 |
| 20 | 18 | 0.0 | 0.0 | 0.1 | 3.3 | 7.7 | 17.8 | 11.7 | 19.4 | 22.0 | 19.7 | 28.2 | 22.9 | 19.6 | 13.6 |
| 20 | 19 | 0.1 | 0.3 | 0.4 | 17.4 | 60.3 | 67.9 | 56.9 | 85.2 | 88.2 | 90.3 | 96.7 | 99.4 | 99.6 | 99.1 |
| 20 | 19 | 0.1 | 0.3 | 0.4 | 17.4 | 60.3 | 67.9 | 56.9 | 85.2 | 88.2 | 90.3 | 96.7 | 99.4 | 99.6 | 99.1 |

Table 10: Models' Performance on the WC Task across Different Task Configurations.

| Setting | | | Pythia Models Accuracy (%) | | | | | | | | | | LLaMA Models Accuracy (%) | | | |
|---|---|---|---|---|---|---|---|---|---|---|---|---|---|---|---|---|
| $|F|$ | $|L|$ | $|D|$ | 14M | 31M | 70M | 160M | 410M | 1B | 1.4B | 2.8B | 6.9B | 12B | 3.2-1B | 3.2-3B | 3.1-8B | 3.1-8B-Inst |
| 1 | 2 | 0 | 5.2 | 12.2 | 17.5 | 51.8 | 95.8 | 96.6 | 93.7 | 95.7 | 99.0 | 99.6 | 99.2 | 98.7 | 100.0 | 99.8 |
| 1 | 2 | 1 | 5.0 | 8.8 | 10.3 | 36.5 | 86.6 | 88.4 | 88.2 | 84.0 | 91.7 | 91.9 | 91.1 | 97.8 | 98.5 | 99.1 |
| 1 | 2 | 5 | 4.5 | 6.8 | 12.9 | 37.0 | 64.3 | 68.5 | 64.5 | 67.3 | 69.5 | 68.9 | 65.3 | 67.1 | 74.2 | 79.8 |
| 1 | 2 | 10 | 6.5 | 10.4 | 17.0 | 34.3 | 59.4 | 61.7 | 61.9 | 56.9 | 60.1 | 60.0 | 53.4 | 53.7 | 63.3 | 63.3 |
| 3 | 2 | 0 | 5.3 | 14.3 | 31.1 | 57.3 | 97.4 | 98.0 | 99.4 | 97.6 | 99.8 | 99.7 | 98.9 | 98.6 | 100.0 | 100.0 |
| 3 | 2 | 1 | 5.7 | 8.8 | 20.6 | 46.7 | 94.5 | 97.9 | 98.0 | 93.2 | 98.8 | 99.1 | 98.9 | 97.9 | 100.0 | 100.0 |
| 3 | 2 | 5 | 5.5 | 8.8 | 20.4 | 40.6 | 80.7 | 83.7 | 82.8 | 78.8 | 87.9 | 88.0 | 84.0 | 75.5 | 95.5 | 97.2 |
| 3 | 2 | 10 | 6.1 | 9.4 | 20.2 | 36.5 | 65.9 | 72.3 | 77.1 | 60.7 | 73.3 | 73.0 | 63.9 | 62.9 | 80.1 | 84.1 |
| 5 | 2 | 0 | 5.0 | 11.0 | 33.9 | 58.1 | 98.2 | 99.4 | 99.2 | 97.2 | 99.6 | 99.8 | 98.3 | 97.2 | 100.0 | 100.0 |
| 5 | 2 | 1 | 6.3 | 11.0 | 27.0 | 52.6 | 95.4 | 98.2 | 98.9 | 94.3 | 99.6 | 99.7 | 97.9 | 95.6 | 100.0 | 100.0 |
| 5 | 2 | 5 | 6.5 | 7.8 | 20.4 | 39.7 | 84.4 | 90.8 | 88.4 | 76.1 | 90.4 | 92.6 | 86.3 | 81.3 | 98.6 | 99.2 |
| 5 | 2 | 10 | 3.6 | 9.0 | 19.0 | 36.5 | 69.3 | 78.6 | 80.5 | 65.2 | 81.5 | 75.9 | 60.3 | 59.6 | 88.8 | 89.4 |
| 1 | 3 | 0 | 2.4 | 9.2 | 18.5 | 41.3 | 89.5 | 91.9 | 88.2 | 94.8 | 99.0 | 99.5 | 99.0 | 98.2 | 99.9 | 99.9 |
| 1 | 3 | 1 | 2.3 | 4.9 | 8.2 | 28.6 | 79.1 | 85.7 | 84.1 | 81.0 | 87.9 | 90.8 | 92.3 | 96.3 | 98.9 | 98.9 |
| 1 | 3 | 5 | 2.2 | 5.2 | 9.4 | 20.6 | 52.1 | 53.2 | 51.3 | 52.1 | 54.1 | 60.2 | 50.0 | 49.9 | 63.6 | 73.2 |
| 1 | 3 | 10 | 2.9 | 4.1 | 12.7 | 21.4 | 45.9 | 48.8 | 45.7 | 41.2 | 45.0 | 43.3 | 39.3 | 38.3 | 44.5 | 49.4 |
| 3 | 3 | 0 | 3.1 | 6.4 | 29.5 | 56.0 | 98.6 | 96.9 | 98.5 | 97.9 | 100.0 | 100.0 | 99.0 | 97.7 | 100.0 | 100.0 |
| 3 | 3 | 1 | 3.0 | 5.2 | 18.6 | 42.3 | 94.6 | 96.8 | 96.9 | 95.4 | 99.4 | 99.6 | 98.2 | 97.2 | 99.9 | 100.0 |
| 3 | 3 | 5 | 3.1 | 3.6 | 17.2 | 28.5 | 73.9 | 75.1 | 77.3 | 71.6 | 79.2 | 83.6 | 78.8 | 65.8 | 97.5 | 98.8 |
| 3 | 3 | 10 | 2.5 | 3.7 | 13.2 | 24.8 | 55.6 | 59.5 | 64.6 | 45.5 | 61.3 | 57.8 | 44.2 | 39.4 | 80.2 | 83.8 |
| 5 | 3 | 0 | 3.3 | 5.6 | 31.5 | 53.2 | 98.2 | 99.8 | 99.7 | 98.3 | 99.8 | 100.0 | 98.3 | 97.6 | 100.0 | 100.0 |
| 5 | 3 | 1 | 2.7 | 3.2 | 27.4 | 44.3 | 95.2 | 98.2 | 98.6 | 95.5 | 99.6 | 99.9 | 98.3 | 96.0 | 100.0 | 100.0 |
| 5 | 3 | 5 | 3.7 | 4.7 | 18.7 | 31.4 | 76.2 | 84.6 | 87.8 | 70.8 | 88.1 | 90.8 | 82.5 | 70.1 | 99.3 | 99.9 |
| 5 | 3 | 10 | 2.4 | 4.1 | 14.8 | 23.4 | 56.5 | 71.8 | 74.9 | 50.5 | 67.5 | 65.0 | 42.3 | 44.0 | 90.5 | 92.5 |
| 1 | 4 | 0 | 1.4 | 6.9 | 17.2 | 40.9 | 85.8 | 85.8 | 85.7 | 94.3 | 99.3 | 99.5 | 98.9 | 98.2 | 100.0 | 100.0 |
| 1 | 4 | 1 | 2.2 | 3.0 | 9.3 | 22.5 | 76.2 | 83.1 | 76.7 | 80.2 | 84.3 | 90.4 | 91.3 | 94.9 | 97.8 | 98.2 |
| 1 | 4 | 5 | 1.6 | 2.8 | 9.8 | 15.5 | 43.3 | 44.2 | 43.0 | 44.7 | 41.2 | 48.2 | 45.7 | 37.8 | 60.5 | 67.8 |
| 1 | 4 | 10 | 1.9 | 2.6 | 8.1 | 12.3 | 38.0 | 39.9 | 37.2 | 32.5 | 34.5 | 38.5 | 30.1 | 26.8 | 41.3 | 44.0 |
| 3 | 4 | 0 | 1.2 | 3.9 | 26.3 | 53.3 | 97.7 | 95.9 | 98.3 | 97.6 | 99.9 | 100.0 | 98.4 | 97.7 | 100.0 | 100.0 |
| 3 | 4 | 1 | 2.2 | 3.0 | 16.1 | 37.1 | 92.7 | 96.0 | 96.4 | 94.4 | 98.8 | 99.2 | 97.5 | 96.6 | 100.0 | 100.0 |
| 3 | 4 | 5 | 3.0 | 2.2 | 14.5 | 22.2 | 70.2 | 67.3 | 72.9 | 61.1 | 75.3 | 82.0 | 74.6 | 50.0 | 97.0 | 97.9 |
| 3 | 4 | 10 | 1.4 | 2.4 | 8.9 | 16.8 | 46.6 | 55.4 | 58.6 | 38.6 | 51.0 | 55.9 | 34.9 | 31.0 | 81.8 | 83.7 |
| 5 | 4 | 0 | 2.2 | 3.3 | 28.6 | 50.2 | 98.4 | 98.6 | 99.7 | 97.7 | 99.9 | 99.8 | 98.0 | 96.2 | 100.0 | 100.0 |
| 5 | 4 | 1 | 2.1 | 4.1 | 21.9 | 38.5 | 96.0 | 97.5 | 98.3 | 96.3 | 99.9 | 99.7 | 98.1 | 95.0 | 100.0 | 100.0 |
| 5 | 4 | 5 | 1.6 | 2.9 | 14.0 | 22.7 | 75.2 | 79.7 | 85.8 | 65.1 | 86.8 | 90.0 | 80.5 | 62.1 | 99.4 | 99.9 |
| 5 | 4 | 10 | 1.9 | 1.9 | 12.3 | 18.9 | 49.0 | 64.8 | 70.0 | 43.3 | 60.5 | 61.3 | 38.7 | 31.4 | 90.4 | 94.6 |
| 1 | 5 | 0 | 1.6 | 4.8 | 17.8 | 41.4 | 84.9 | 80.3 | 83.3 | 93.2 | 97.9 | 98.4 | 98.7 | 97.4 | 99.6 | 99.5 |
| 1 | 5 | 1 | 1.2 | 2.4 | 8.0 | 20.5 | 66.1 | 75.2 | 72.5 | 75.4 | 80.8 | 87.4 | 88.4 | 93.9 | 97.9 | 98.2 |
| 1 | 5 | 5 | 1.4 | 2.5 | 7.7 | 11.0 | 38.2 | 38.0 | 35.6 | 38.5 | 33.3 | 42.3 | 38.4 | 24.3 | 56.0 | 62.8 |
| 1 | 5 | 10 | 1.3 | 1.0 | 4.8 | 9.9 | 29.9 | 32.5 | 31.5 | 31.4 | 30.0 | 31.6 | 24.5 | 18.9 | 35.2 | 38.1 |
| 3 | 5 | 0 | 1.1 | 3.2 | 28.7 | 52.6 | 96.1 | 93.3 | 97.9 | 98.8 | 99.6 | 100.0 | 97.9 | 96.7 | 100.0 | 100.0 |
| 3 | 5 | 1 | 1.5 | 2.8 | 18.2 | 39.4 | 89.0 | 95.1 | 95.6 | 93.6 | 99.4 | 99.7 | 97.4 | 95.4 | 99.9 | 100.0 |
| 3 | 5 | 5 | 1.7 | 2.1 | 10.2 | 20.0 | 65.1 | 69.2 | 69.9 | 61.6 | 70.6 | 78.4 | 73.4 | 41.1 | 96.6 | 99.0 |
| 3 | 5 | 10 | 1.5 | 2.3 | 5.0 | 15.7 | 42.1 | 51.3 | 53.3 | 32.6 | 45.2 | 51.1 | 30.1 | 20.9 | 81.6 | 84.4 |
| 5 | 5 | 0 | 1.6 | 2.5 | 25.7 | 48.8 | 98.2 | 98.4 | 99.4 | 98.7 | 99.9 | 100.0 | 97.9 | 94.7 | 100.0 | 100.0 |
| 5 | 5 | 1 | 1.6 | 3.0 | 19.5 | 38.7 | 96.0 | 97.5 | 98.6 | 95.9 | 99.7 | 99.9 | 98.2 | 94.6 | 100.0 | 100.0 |
| 5 | 5 | 5 | 1.2 | 2.2 | 8.4 | 19.9 | 70.5 | 81.3 | 83.6 | 60.4 | 82.4 | 88.6 | 79.7 | 55.3 | 99.7 | 99.9 |
| 5 | 5 | 10 | 1.1 | 2.0 | 7.2 | 14.2 | 45.8 | 64.8 | 65.5 | 33.8 | 55.4 | 55.2 | 30.7 | 25.6 | 92.7 | 95.6 |

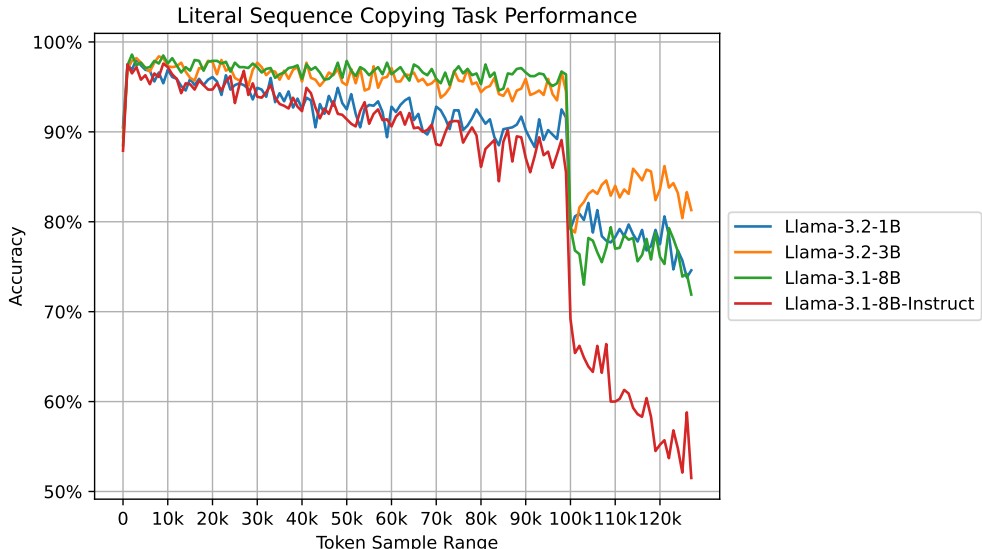

Figure 11: LM performance on the LSC task deteriorates as tokens are sampled from ranges with larger indices for Llama models as well. A clear drop is observed at 100k, where Llama's tokenizer switches from `tiktoken` tokens to the 28K additional non-English rare tokens. Furthermore, within the regular tokens (first 100k), there is a clear decreasing trend.

Table 11: Pearson correlation between the models' LSC performance and tokenizer index for the first 100k regular tokens. This result confirms that the decreasing trend across the first 100k Llama tokens is statistically significant according to the Pearson correlation test.

|  | Llama-3.2-1B | Llama-3.2-3B | Llama-3.1-8B | Llama-3.1-8B-Instruct |
|---|---|---|---|---|
| Pearson $r$ | -0.8933 | -0.7396 | -0.6263 | -0.9382 |
| $p$-value | 1.8325e-35 | 2.2564e-18 | 4.1340e-12 | 1.7243e-46 |

## C Supplementary Results I: ICL Performance Strongly Correlates with Token Frequency − Llama Models

In Section 4.2, we showed that LMs' ICL performance correlates with token frequency across Pythia models. Here, we conduct the same experiments for Llama models. The tokenizer used in the Llama LM family differs from that of Pythia, as it combines 100K tokens from the `tiktoken`[16] tokenizer with 28K additional rare, non-English tokens to better support non-English languages (Grattafiori et al., 2024). Figure 11 shows the LSC performance of various Llama models across tokenizer index ranges. As with Pythia, we observe a clear decreasing trend among the regular tokens (the first 100K), with a drastic dip occurring exactly at the 100K mark where the additional rare tokens begin. Table 11 confirms that this decreasing trend is statistically significant. These results show that our finding is not unique to Pythia and that it holds across LM families, suggesting it is more likely an inherent property of ICL.

---

[16]https://github.com/openai/tiktoken/tree/main

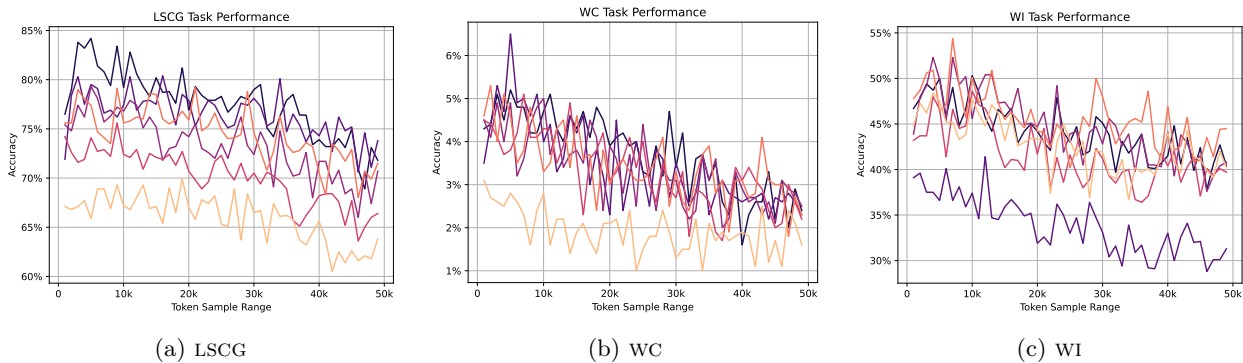

(a) LSCG         (b) WC         (c) WI

Figure 12: LM performance on other literal-based tasks (LSCG, WC, and WI) also deteriorates as tokens are sampled from ranges with larger indices. Line plot colours correspond to model size, following the same colour scheme as Figure 1.

Table 12: Pearson correlation between the models' LSC performance and tokenizer index for the first 100k regular tokens. This result confirms that the decreasing trend across the first 100k Llama tokens is statistically significant according to the Pearson correlation test.

| Model Size | 410m | 1b | 1.4b | 2.8b | 6.9b | 12b |
|---|---|---|---|---|---|---|
| Task: LSCG | | | | | | |
| Pearson $r$ | -0.742 | -0.731 | -0.876 | -0.842 | -0.588 | -0.840 |
| $p$-value | 1.040E-09 | 2.531E-09 | 1.829E-16 | 3.401E-14 | 8.947E-06 | 4.357E-14 |
| Task: WC | | | | | | |
| Pearson $r$ | -0.501 | -0.701 | -0.763 | -0.736 | -0.775 | -0.852 |
| $p$-value | 2.482E-04 | 2.010E-08 | 1.931E-10 | 1.623E-09 | 6.412E-11 | 8.612E-15 |
| Task: WI | | | | | | |
| Pearson $r$ | -0.716 | -0.639 | -0.684 | -0.814 | -0.820 | -0.805 |
| $p$-value | 7.119E-09 | 7.829E-07 | 6.136E-08 | 1.199E-12 | 5.551E-13 | 3.043E-12 |

# D   Supplementary Results II: ICL Performance Strongly Correlates with Token Frequency Across Tasks

In this section, we list the token frequency experiments for other literal-based tasks. As shown in Figure 12 and Table 12, we see that the same finding persists: ICL performance strongly correlates with token frequency across tasks.

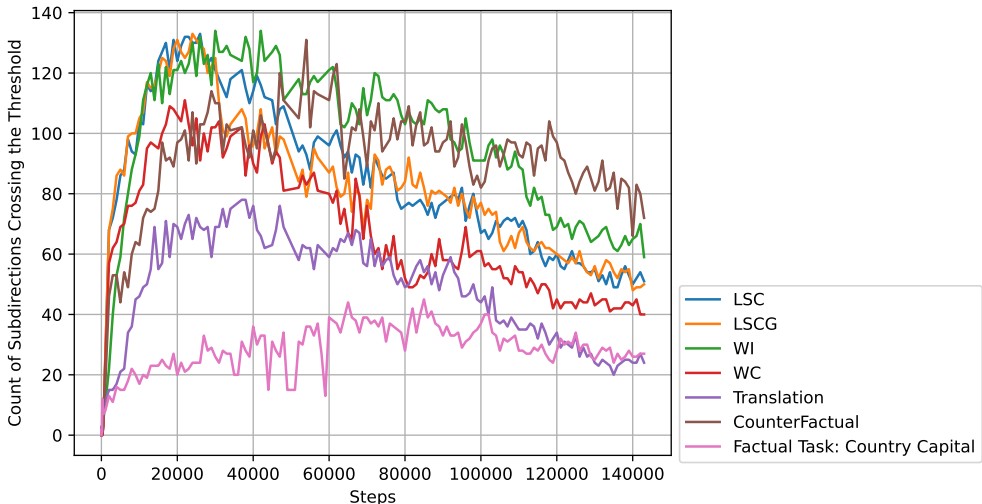

Figure 13: Threshold dynamics of singular value directions in the CITYCAPITAL task, showing a distinct trajectory compared to ICL tasks.

## E  SUDA Analysis for Factual Tasks

To examine whether the observed subspace formation trend generalises beyond ICL, we applied the same SUDA analysis to a factual recall task, CITYCAPITAL, which requires retrieving the capital city given a country name. Interestingly, the formation of singular value directions in this task follows a distinctly different trajectory from that observed in the ICL tasks, with no clear rise of the dominant components that characterise in-context learning. This suggests that the progressive alignment of singular directions may be a property specific to ICL rather than a universal feature of model adaptation. A more systematic investigation across diverse factual and reasoning tasks is left for future work.

