# OpenReview forum: "Illusion or Algorithm? Investigating Memorization, Emergence, and Symbolic Processing in In-Context Learning"
_TMLR — Accepted by TMLR_

### Review · Reviewer_VkLq · 2025-06-08

**Summary Of Contributions:**

This paper introduces a novel analytical framework that unifies and explains the diverse approaches and results of previous research on the phenomenon known as in-context learning (ICL). Specifically, the authors introduce a suite of synthetic tasks (LSC, LSGC, WI, and WC) designed to assess whether models can correctly generalize patterns they have not memorized. The paper further proposes two additional tasks (TT and CF) that require understanding more complex, realistic scenarios. These methodologies are developed to show that ICL does not simply rely on memorization or matching of known patterns. Empirical results show that (1) for models above a certain size, ICL performance becomes predictable across these tasks, (2) models learn to perform tasks including frequent tokens more quickly and accurately, and (3) these trends persist even at intermediate stages of training. The authors also conduct a mechanical analysis by measuring the effective dimension of subspaces in the residual stream and establishing correlations between this and ICL competence.

**Audience:**

Yes

**Broader Impact Concerns:**

As this paper is an analytical study aimed at deepening our understanding of ICL mechanisms, I don’t think this paper needs further ethical statements. The discussion section already provides sufficient reflection on broader implications.

**Claims And Evidence:**

Yes

**Requested Changes:**

Please see the weaknesses above. I look forward to further development of these points through discussion.

**Strengths And Weaknesses:**

**[Strengths]**

* The paper introduces a diverse set of new tasks, explores multiple model sizes, and evaluates performance at various training checkpoints. This breadth of analysis offers a new perspective on ICL phenomena.
* The token frequency-based analysis has not been explored much in the literature; demonstrating that ICL occurs more robustly for frequent tokens is a significant finding with broad implications for both current and future research.
* Comparative analyses with prior work are thorough, clearly highlighting the contributions of this study. In particular, the authors suggest conflicting results in the literature would be the effect of new findings, such as token frequency.

**[Weaknesses]**

* The settings for the synthetic tasks (LSC, LSGC, WI, WC) may be too short; it would be valuable to extend input lengths until even larger models (e.g., 12B parameters) begin to fail, thereby probing the true limits of ICL. As these tasks are intended as evaluations of ICL, a deeper analysis of ICL’s limitations is encouraged. Furthermore, as context lengths grow (to the order of a few hundred tokens), the possibilities of pattern matching could be reduced substantially.
* Concerns regarding the token position remain. Although LLMs employ relative positional encodings, phenomena such as attention sinks suggest that absolute positions can affect model behavior. Since ICL is expected to operate over long contexts in practice, it would be important to analyze the impact of token position.
* Including TT and CF results in Table 1 would show the evaluation more comprehensively.
* Additional explanation or analysis of instruction tuning is needed. For instance, Llama 3.1-8B has both instruction-tuned and pre-trained results in Table 1, but the mixed results make interpretation difficult; similarly, Llama 3.2-1B/3B also have instruction-tuned versions that should be reported. In addition, instruction-tuned models are likely to have an advantage on the CF task.
* While the subspace-direction analysis is interesting and new, its connection to ICL remains questionable. One could argue that a dominant subspace direction learning more frequent tokens first is an expected outcome; a more concrete discussion of how this mechanistic finding connects to ICL would strengthen the work.

---

> ### Author Response · Authors · 2025-07-28
> **Author Response**
>
> We thank the reviewer for the thorough and positive review. We are particularly encouraged by your recognition of the work's contributions including the novelty of the frequency analysis experimental design. Thank you also for the very insightful suggestions, we have detailed our response to your questions below:
>
> **Q1: Stretching the difficulty**
>
> Thank you for this suggestion. We did experiment with more complex settings until the 12B model failed. For all of LSCG, WI, and WC, we have seen Pythia-12B, and even LLaMA 3.1, reach performance levels around 60% or lower. Please see the entries in Appendix B. The only exception is LSC, where the configuration is too simple to yield any cases where the model fails. However, LSCG can be seen as a more difficult variant of LSC.
>
> That said, we agree that evaluating each model until "breaking point" is a very interesting direction for future work and would enable us to test the ICL capability limits. Our focus was on establishing the nature of ICL (its dependence on token frequency and task configuration).
>
> **Q2: Token Position**
>
> Thank you for highlighting this insightful point on the potential effect of positional embeddings. While we do not explicitly control for this, our current experimental design with shorter context lengths likely mitigates the impact of variables. Also, one of the motivations for the design of the WI task is to take the relative token position into account (although this is relative position and not absolute position within a much larger context).
>
> We agree that this is a very interesting avenue of potential future work especially combined with the suggestion to test longer contexts.
>
> **Q3: Token translation & counterfactual results**
>
> Thank you for pointing out this oversight on our part. We have now included the token translation and counterfactual results in the revised Table 2 and Appendix B. Please refer to the updated manuscript for the new tables.
>
> (We believe you may be referring to a results summary table, as Table 1 describes our task setups)
>
> **Q4: Discussion around instruction fine-tuning**
>
> Thank you for highlighting the potential impact of instruction fine-tuning on some of the tasks we use. We agree that the difference in the mechanisms of LMs compared to their IFT’ed counterparts is very interesting; however, our focus was in exploring the nature of ICL on pre-trained models and the inclusion of instruction-tuned models was to ensure completeness. A complete analysis of the effect of IFT is beyond the scope of this work and is better suited for immediate follow-up research. Moreover, the IFT process used by the Llama family, and other open-weight LM families, is not entirely opaque. This remains a promising direction, especially as projects such as Olmo-2 make further investigation possible.
>
> Additional Experiment: As suggested, we have added the results of the IFT versions of Llama-3.2 1B and 3B in the relevant sections. While a detailed, more controlled analysis is beyond the scope of this work, we hope these results can inspire interesting future research.
>
> **Q5: SUDA**
>
> Thank you for pointing out the need to make this connection more explicit.
>
> Our finding isn't only that the frequent tokens are learned first. It is that the development trajectory of the subspace allocation (number of directions represented in Figure 10b) is stable and predictable across all ICL tasks. This suggests that the different tasks are "drawing on" the same underlying ICL mechanism and this mechanism first generalises (uses many directions) and then specialises (becomes more efficient).
>
> To further validate the relation between ICL and the discovered subspaces, we add two additional experiments in the revised draft, as follows:
>
> - *What happens if we analyse a non-ICL task?*
> We introduce a simple factual recall task (CountryCapital). We observe that the formation of the answer-containing subspace is distinctly different compared to the ICL task.
> - *How do different tasks share the answer containing subspace?*
> We observe that 1) tasks that only require pattern-matching from context (LSC, LSCG, WI, and WC) show a high degree of subspace-sharing, 2) tasks that require usage of pretrained knowledge use distinctly separate subspaces.
>
> Taken together, our analysis establishes a stronger relation between the answer-containing subspaces, their formation over training, and the type of task (i.e., pure in-context pattern matching, mixture of in-context pattern matching and pretrained knowledge, and pure pretrained knowledge).
>
> We have revised the discussion in Section 6.2 to make this connection explicit.

---

> > ### Comment · Reviewer_VkLq · 2025-08-03
> >
> > Thank you for the response, clarification, and additional experiments. In particular, experiments using instruction-tuned models and new tasks/analyses seem to improve the overall quality. The finding is interesting that the model behaves differently between ICL and non-ICL tasks, and even for different tasks, ICL tasks share a considerable amount of subspaces.

---

> > > ### Author Response · Authors · 2025-08-04
> > >
> > > Thank you for the response. We appreciate your interest in our work.

---

### Review · Reviewer_vops · 2025-06-15

**Summary Of Contributions:**

This paper investigates whether in-context learning (ICL) in language models stems from memorization, symbolic algorithm execution, or genuine generalization. Using a suite of diagnostic tasks and models from the Pythia scaling suite, the authors find that while ICL goes beyond simple memorization, it is not equivalent to symbolic algorithmic reasoning. Instead, ICL competence develops gradually, depends on token frequency and task configuration, and aligns with predictable trends during pretraining. The work provides nuanced insights into how ICL emerges and its implications for AI safety and interpretability.

**Audience:**

Yes

**Claims And Evidence:**

Yes

**Requested Changes:**

Please make the required changes to address the concerns raised as the weaknesses.

**Strengths And Weaknesses:**

Strengths:
1. The work presents a comprehensive set of experiments, not only on fully trained models but also on intermediate checkpoints throughout the pretraining process.
2. It introduces a novel idea of using a token’s index as a proxy for its frequency in the training data.
3. It proposes that ICL capability develops gradually with model size, rather than emerging abruptly, a valuable insight for AI safety, particularly in estimating the safety implications of models of a given size.

Weaknesses:
1. Although the paper introduces the novel idea of using token index as a proxy for token frequency in the pretraining data, most experiments are conducted only on the Pythia model, whose training data is publicly available. It’s unclear why the authors opted for a proxy instead of using the actual data, especially when it’s unknown how good token indices are as the proxy.
2. Some experiments are conducted on only a single dataset, while others are performed across all datasets. For example, it’s unclear why the experiment in Section 4.2 was not run on all datasets.
3. The contribution of the results in Section 4.3 to the central question regarding the nature of the ICL mechanism, i.e., whether it is based on memorization or symbolic algorithms, is not clearly explained.
4. The validity of the *Singular Residual Stream Direction Analysis* method is questionable. Since singular vectors are computed using the residual stream at the last layer (which likely contains information about the answer token), it’s not surprising that some singular vectors align with the answer token. This alignment may not indicate that the vector "performs" the task.
    - To convincingly demonstrate that a singular vector can perform an ICL task, the authors should conduct a causal intervention experiment. Specifically, they should show that intervening along the singular vector direction in the residual stream changes the model’s output from an incorrect token to the correct one.
6. The presentation of the paper could be improved in the following ways:
    - In Section 1, the second sentence redundantly redefines Language Models (LMs), which is already done in the first sentence.
    - On page 2, in the paragraph titled Finding 1, “describ” should be corrected to “describe.”
    - The first paragraph of Section 3 defines ICL. However, I’m not fully sure if I understood it properly. It would be helpful to also include examples of tasks that do not qualify as ICL under this definition.
    - The Word Content task is difficult to understand as currently phrased. Rewriting it for clarity would improve comprehension.
    - Across all figures, increase the font size of text elements (e.g., axis labels, titles, legends) for better readability.

---

> ### Author Response · Authors · 2025-07-28
> **Author Response**
>
> Thank you for your detailed and constructive feedback. We found the detailed comments to be very valuable.
> Based on your feedback, we have:
> - Performed new experiments to strengthen our findings, and
> - Revised key sections to clarify our methodology and reasoning.
>
> We believe these changes directly address the reviewers’ comments and strengthen the manuscript.
>
> **Q1: Questions regarding tokenizer index**
>
> Thank you for pointing out this assumption. We performed additional experiments to verify the effectiveness of our methods.
>
> We use the tokenizer index as a proxy for two reasons. First, this method is applicable to both models with open and closed training datasets. Particularly, we also experimented with Llama models, where the dataset has not been released. This approach allowed us to assess the effect across both models simultaneously. Second, our method is significantly more efficient. The Pile dataset contains 825 GiB of text. Processing it in full to extract token-level statistics would be computationally expensive and time-consuming.
>
> That said, we agree with the reviewer that confirming the validity is beneficial. We sampled 5,000 tokens from the Pythia tokenizer, counted their token frequencies on 250,000 documents from the Pile, and computed the correlation between tokenizer index and token count. We found a strong and significant correlation, which confirms the quality of our approach.
> - Spearman r = -0.746, p<1e-100
> - Pearson r = -0.701, p<1e-100
>
> Again, thank you for your attention to detail. We have added a brief discussion in Section 4.2.
>
> **Q2: Token index experiments on other task**
>
> Thank you for this excellent suggestion!
>
> Based on your feedback, we have extended the analysis to the other three literal-based tasks: LSCG, WC, and WI, and observed the same trend. Please see Section 4.2 and Appendix D for details.
>
> **Q3: Position on the memorization vs algorithm debate**
>
> We'd like to thank the reviewer for highlighting that this connection was not made sufficiently clear.
>
> Our argument is as follows: A true "symbolic algorithm" should be robust to minor changes to the task structure. However, we find that task performance is related to:
> - The gap length in the LSCG task.
> - The sequence length in the WI task.
> - The number of distractors or features in the WC task.
>
> This effect demonstrates that model capability is not similar to an "abstract algorithm" but instead reliant on underlying statistical information. This is a novel and significant finding and supports our central position that ICL is a "peculiar mixture of generalization and dependence on token statistics."
>
> **Q4: SUDA & causal analysis**
>
> Thank you for highlighting this important aspect of our methodology. We’d like to point out a key detail that we believe addresses the concern.
>
> The SVD decomposition is performed on the **unembedding** matrix, not the residual stream. The resulting orthogonal matrix that is used to map the residual stream to the singular vectors, therefore, does not have any connection to the answer token.
>
> That said, our SUDA analysis presents an interesting result not because it identifies the presence of an “answer-containing subspace,” but because SUDA can enumerate the answer-containing subspace. By studying their evolution over training, we observe a common trend of initial increase followed by gradual decline, which we hypothesise corresponds to specialisation forming within the model.
>
> To enforce our point, we add two additional results in Section 6:
> - *What happens if we analyse a non-ICL task?* We introduce a simple factual recall task (CountryCapital). We observe that the formation of the answer-containing subspace is distinctly different compared to the ICL task.
> - *How do different tasks share the answer containing subspace?* We observe that 1) tasks that only require pattern-matching from context (LSC, LSCG, WI, and WC) show a high degree of subspace-sharing, 2) tasks that require usage of pretrained knowledge use distinctly separate subspaces.
>
> Since the attention heads are low rank (each attention head performs operations on the hidden state using a fraction of the model dimension), placing the answer token consistently over a certain subspace is possible if and only if there are certain attention heads that specialize in these tasks. A different answer-containing subspace for a task signals the presence of a different mechanism.
>
> We do agree with the reviewer that a causal mediation analysis in the subspace-level can inform a more in-depth understanding of the exact causal process the model uses. However, our results with SUDA are enough to support our claim that: 1) model specializes on in-context pattern learning as the training progresses in a smooth, predictable manner, 2) incorporation of pretrained knowledge happens in a separate mechanism that evolves differently.
>
> **Q5: Presentation and Typos**
>
> Thank you for the helpful feedback! We've revised the manuscript accordingly.

---

> > ### Comment · Reviewer_vops · 2025-08-02
> >
> > Dear Authors,
> >
> > Thank you for addressing my concerns in the revised version and for clarifying Section 4.3 and the SUDA technique. I recommend including your perspective on the memorization vs. algorithm debate at the end of the section 4.3. Additionally, consider citing more recent works on this topic [1–5].
> >
> > Aside from that, I’m satisfied with your responses and believe this is a well-rounded paper, offering both interesting insights and methods that would be valuable to the community.
> >
> >
> > [1] Prakash et al, “Language Models use Lookbacks to Track Beliefs”, 2025.
> >
> > [2] Yang et al, “Emergent Symbolic Mechanisms Support Abstract Reasoning in Large Language Models”, 2025.
> >
> > [3] Wu et al, “How Do Transformers Learn Variable Binding in Symbolic Programs?”, 2025.
> >
> > [4] Shojaee et al, “The Illusion of Thinking: Understanding the Strengths and Limitations of Reasoning Models via the Lens of Problem Complexity”, 2025.
> >
> > [5] Mirzadeh et al, “GSM-Symbolic: Understanding the Limitations of Mathematical Reasoning in Large Language Models” 2024.

---

> > > ### Author Response · Authors · 2025-08-04
> > >
> > > Thank you for the great suggestions on recent work. They’re all highly relevant and really help bring the paper up to date. Based on your constructive feedback, we’ve (1) extended our reasoning at the end of Section 4.3 and (2) incorporated the newer related work throughout the paper. We truly appreciate your thoughtful input.

---

### Review · Reviewer_4mNh · 2025-07-18

**Summary Of Contributions:**

This paper presents a very interesting and thorough analysis of how the in-context learning (ICL) capability emerges and improves with the scaling of the model size, and with more pre-training steps. In the literature, there has been different opinions on whether LLM in-context learning is due to pure memorization, or the model really discovers the underlying symbolic algorithm underneath the in-context examples. To understand the discrepancy of conclusions in existing studies, the authors introduce a suite of synthetic tasks, including several token prediction tasks that do not require any prior knowledge of natural language semantic meanings, as well as token translation and counterfactual tasks. They show that the ICL performance improvement is sharper w.r.t. the model scale on token prediction tasks, while the improvement is gradual and predictable by a scaling law on token translation. In particular, the study reveals that ICL performance is lower on rarer tokens than common tokens, indicating that the model is not learning a robust symbolic algorithm despite good generalization performance beyond pure memorization.

**Audience:**

Yes

**Broader Impact Concerns:**

No concern.

**Claims And Evidence:**

Yes

**Requested Changes:**

1. Please present examples of the prompt provided to the model. Does the prompt only contain in-context examples, or does it also include some high-level instructions of the task?

2. In Table 2 and 6, the performance does not necessarily increase with model scaling. Where does this variance come from? For example, if the authors generate a larger eval set and sample different subsets of 1000 examples, how does the performance variance look like?

3. The model scale studies in this work is overall a bit small. Have the authors conducted this study on larger models? There are more recent opensource model families with larger models, such as LLaMA and Qwen.

4. I did not see token translation and counterfactual task performance in Table 2. Is there a similar summary of results on these tasks for both Pythia and LLaMA models?

5. In Figure 2 (b), it is interesting to see that index i = 1 consistently achieves the worst performance for different sequence lengths. Intuitively, the bottom of the U-shape trends should be around the middle. Any intuition about this observation?

6. In Figure 10 (b), any intuition on why the trend is a decline in the later training stage?

**Strengths And Weaknesses:**

Strengths:

In general I enjoy reading this paper, it provides some interesting and novel insights, and the empirical study is very thorough.

1. The task suite is well-designed. It covers tasks of a wide range of difficulty, and includes both tasks that require and do not require natural language prior knowledge. It is a good test bed to evaluate ICL performance that does not come from pure memorization, as the tasks are synthetic and the exact token sequences should not appear in pretraining.

2. The analysis over different token index ranges is interesting and well-motivated. Intuitively, it is reasonable to imagine that LLMs perform better than common tokens than rare tokens, and the learning progress on common tokens is faster than rare tokens. It is nice to see a rigorous study to confirm these hypotheses.

3. The SUDA analysis is also interesting. It is intriguing to see that the training steps with peak maximum logit values align across different tasks, and they align with the initial training steps with significant performance leap in Section 5.


Weaknesses:

I have some clarification questions below.

1. Can the authors present examples of the prompt provided to the model? Does the prompt only contain in-context examples, or does it also include some high-level instructions of the task?

2. In Table 2 and 6, the performance does not necessarily increase with model scaling. Where does this variance come from? For example, if the authors generate a larger eval set and sample different subsets of 1000 examples, how does the performance variance look like?

3. The model scale studies in this work is overall a bit small. Have the authors conducted this study on larger models? There are more recent opensource model families with larger models, such as LLaMA and Qwen.

4. I did not see token translation and counterfactual task performance in Table 2. Is there a similar summary of results on these tasks for both Pythia and LLaMA models?

5. In Figure 2 (b), it is interesting to see that index i = 1 consistently achieves the worst performance for different sequence lengths. Intuitively, the bottom of the U-shape trends should be around the middle. Any intuition about this observation?

6. In Figure 10 (b), any intuition on why the trend is a decline in the later training stage?

---

> ### Author Response · Authors · 2025-07-28
> **Author Response**
>
> We would like to thank the reviewer for the encouragement and the overwhelmingly positive review! We are truly grateful and would also like to thank the reviewer for the thoughtful questions, which we have answered below:
>
> **Q1: Examples of prompt settings.**
>
> Our prompts were predominantly without explicit instructions. This was intentional because we aimed to control the experiment to focus solely on ICL. The only exception is the counterfactual (CF) task, where instructions are necessary to elicit meaningful performance. We provide examples of our prompts in Table 1. This table provides a "template" and a concrete example for each of the tasks.
>
> We provide a few example prompts below. An anonymous repository containing the full prompt generation code and additional examples is also available in the paper. Please refer to the repository or Table 1 for more examples.
>
> - **LSC**: ' Eyes rest voice folding Koch generalize Aviation Eyes rest voice'
> - **LSCG**: ' logistics boasts pan acquired shouting negotiation substituting rebellion observing wheels logistics boasts pan toy override negotiation'
> - **WC**: ' Koch castle pressures could negotiation Copenhagen catalysts castle extensive\n soluble\n … canvas\n acquired wheels surveyed Eyes Eyes assassination castle Copenhagen pressures\n',
> - **WI**: ' folding Eyes rest -> rest; voice Koch generalize -> generalize; Aviation stator Except -> Except; castle deposition Approximately -> Approximately; substituting logistics boasts -> boasts; pan rebellion observing ->',
> - **Translate**: ' friend -> Freund; white -> Weiss; mandarin -> Mandarine; knife -> Messer; chicken -> Huhn; eight ->'
> - **Counterfactual**: 'If we switch the capital of Austria and Belgium, then Austria's capital is Brussels and Belgium's capital is'
>
> **Q2: Variance in scaling trends.**
>
> Thank you for this question, however, we'd like to point out that we do see a general trend that task performance increases with model scaling across the board. The minor variations are likely due to training instability. The only exceptions are tasks that are either too easy or too hard to show any meaningful difference in performance. Could you please point us to some entries to clarify?
>
> **Q3: More model options and sizes.**
>
> Thank you for this suggestion. While it would be valuable to experiment with a larger number of models, our choice of the Pythia suite was not arbitrary. The Pythia suite provides interim checkpoints across the entire pre-training process which enables us to evaluate model performance at a granular level across training steps. I.e., it provides us a unique opportunity to study the pre-training dynamics. On the other hand, the suggested models, Llama and Qwen, have not released their interim checkpoints.
>
> We'd like to highlight that we also experimented with LLaMA 3.1 8B, which was intentionally selected as a more powerful and recent model and also ensured that our findings generalise beyond a single model family.
>
> In short, we do have results from a sufficiently diverse set of LMs (70M~12B Pythia across training checkpoints, as well as different Llama models), and these models are large and capable enough to support our arguments. While additional experiments on more models would be a welcome bonus, nothing matches Pythia in terms of coverage and suitability for our purpose of studying training dynamics.
>
> **Q4: Token translation & counterfactual results.**
>
> Thank you very much for pointing out this oversight on our part. We have now included the token translation and counterfactual results in the revised Table 2. Please refer to the updated manuscript for the new tables.
>
> **Q5: Intuition on WI performance trend.**
>
> Thank you for highlighting this very interesting and intriguing trend! Your intuition is absolutely right in that the model performance is highest at either end.
>
> We believe this is an indicator that the position embedding and input order plays a crucial role in performing the task. The further it is the token to generation (i.e., the last token in prompt), the harder it is for the model to recognise the pattern and perform the task. This also agrees with the observation that longer pattern length $L$ is harder for the model to identify.
>
> **Q6: Intuition on number of subdirection number decrease**
>
> We hypothesise that this may be a sign of circuit specialisation (and increased efficiency) developing in the model with increased training/data, as discussed in Section 6.2. Figure 10b shows the number of subdirections that output logits greater than a threshold $\tau = 0.2$. In other words, the trend reflects how many subdirections “carry” the ability to perform the task. Therefore, the decline may indicate that the model is "specialising": it is using fewer subdirections to perform the same task.

---

### Decision · Action_Editor_GMVJ · 2025-09-06

**Recommendation:** Accept as is

**Audience:**

Yes

**Audience Explanation:**

The paper covers an important topic in the community, understanding in-context learning behavior, providing new insights that most people in the community will find of interest.

**Claims And Evidence:**

Yes

**Claims Explanation:**

The paper provides (as highlighted by reviewers) thorough experimentation, that carefully isolate the explored behavior, providing convincing evidence for the hypothesis they propose. During the rebuttal the authors run additional experiments that answered the worries the reviewers had regarding the results.

Overall, reading the reviews, rebuttal and looking at the paper, I believe the work provides sufficient evidence and novel insights of interest to a wide part of the community.